# Wiedemann–Franz behavior at the Weyl points in compressively strained HgTe

Abu Alex Aravindnath [1,2,3] ✉, Yi-Ju Ho [1,2], Fabian Schmitt [1,2], Dongyun Chen[1,2], Johannes Kleinlein [1,2], Wouter Beugeling [1,2], Hartmut Buhmann [1,2], Stanislau U. Piatrusha [1,2] ✉ & Laurens W. Molenkamp [1,2,3] ✉

Weyl semimetals, with their unique electronic band structure, have drawn significant interest for their potential to explore quantum anomalies in condensed matter systems. In this study, we investigate the large positive magneto-thermal conductance associated with the gravitational anomaly – one of the predicted anomalies – for a Weyl semimetal based on a compressively strained HgTe layer. We clearly identify the Weyl regime in our device and accurately extract the thermal conductance by performing thermometry measurements at liquid helium temperatures using fully electronic methods. We observe the anticipated increase in thermal conductance, and it perfectly matches the electrical conductance according to the Wiedemann–Franz law. This finding indicates that, despite the unique electronic spectrum of Weyl semimetals, the mechanism governing heat transport in this system is the same as that for electrical transport, with no additional violations of conservation laws.

The ability of electrons to carry heat provides a unique approach for exploring mesoscopic systems. Hence, significant interest has been recently attracted to the electron thermal transport in Weyl semimetals, characterized by having their electronic excitations governed by the Weyl equations[1]. These equations feature a linear band dispersion with relativistic electrodynamic properties, reminiscent of the ones encountered in particle physics[2]. This has led to Weyl semimetals being proposed as a condensed matter platform for studying phenomena analogous to those predicted in high-energy physics, such as quantum anomalies. In particular, for Weyl semimetals an increase of electrical conductance with magnetic field has been connected to the chiral anomaly[3,4], while a positive magneto-thermal conductance has been predicted[5,6] and attributed to the gravitational anomaly leading to the violation of separate conservation laws for energy-momentum tensors within chiral Weyl cones.

However, several challenges complicate the observation of these anomalies. One significant obstacle is that the Fermi level of available materials often lies far from the Weyl point, making it difficult to attribute positive magneto-conductance exclusively to the chiral anomaly, as it can also arise from various other mechanisms[7,8]. Observing the gravitational anomaly is even more challenging because heat, unlike charge, is not conserved and can easily dissipate from the electron subsystem into the crystal lattice[9]. Earlier experiments[10–12] have been carried out at relatively high lattice temperatures, where the electron transport is significantly influenced by interaction with phonons, leading to phonon drag and rapid electron-phonon relaxation. Furthermore, exploring thermoelectric phenomena requires using the correct set of constitutive equations[13,14] that reflect the experimental observables (voltage and heat flow). In contrast, the sets employed in refs. 10 and [11] do not meet this criterion. The issue is particularly evident in ref. 11, where the equations chosen lead to an incorrect sign of thermopower correction to the heat conductance. The constitutive equations for thermoelectric coefficients are often derived without directly referencing experimentally observable quantities, notably as in ref. 15, which is widely regarded as a primary reference in this field. However, it is essential to formulate these coefficients using directly measurable quantities to accurately capture their physical meaning. An experiment cannot measure short-circuited thermal conductance, and

[1]Physikalisches Institut (EP3), Universität Würzburg, Würzburg, Germany. [2]Institute for Topological Insulators, Universität Würzburg, Würzburg, Germany. [3]Max Planck Institute for Chemical Physics of Solids, Dresden, Germany. ✉e-mail: abu-alex.aravindnath@physik.uni-wuerzburg.de; stanislau.piatrusha@physik.uni-wuerzburg.de; laurens.molenkamp@physik.uni-wuerzburg.de

in an open circuit, there is always a mixture of the thermopower to the thermal conductance[14].

Compressively strained (≈0.3%) HgTe is a well-established Dirac semimetal[16], in which bulk inversion asymmetry leads to a splitting of the Dirac point into four Weyl points[16,17]. In zero magnetic field, this splitting is small, resulting in transport behavior characteristic of a Dirac semimetal. An in-plane magnetic field breaks the time-reversal symmetry and induces a larger separation of the Weyl nodes in momentum space[18–20], similar to how Dirac nodes split into Weyl nodes in other non-magnetic Dirac/Weyl semimetals, such as $Cd_3As_2$[21] and $Na_3Bi$[22]. This leads to transport signatures typical of Weyl semimetals[16,21,22].

We previously have studied the chiral anomaly, a negative magnetoresistance at the Weyl points in a compressively strained 66 nm thick HgTe layer[16], grown by molecular beam epitaxy (MBE)[23]. These layers have a low intrinsic carrier density and the Fermi level can be tuned (by means of a gate voltage) from the valence band into the conduction band, enabling direct access to the Weyl points, massive and massless (i.e., topological) Volkov–Pankratov surface states, and bulk bands within the same device. In the present paper, we use this ability to unequivocally identify the Weyl regime and address the gravitational anomaly via thermal conductance measurements. These transport phenomena are fundamentally due to the linear dispersion near the Weyl points, as illustrated by the band structure in Fig. 1 for compressively strained HgTe (calculated using $\mathbf{k} \cdot \mathbf{p}$ theory[24]).

In Fig. 1b, we highlight the surface states (shown in maroon) that emanate from the Weyl nodes. The presence of topological surface states near the band crossings is a common property of Dirac and Weyl semimetals[25–27]. Only when the band crossings happen at high symmetry points of the Brillouin zone the surface states are absent. In the presence of a perpendicular magnetic field, these surface states can contribute to quantum oscillations and even give rise to quantum Hall plateaus[16], which complicates an interpretation of magnetoresistance features as signatures of Weyl fermions, using the methods proposed in refs. 28,29.

To detect the electronic thermal conductance exclusively and avoid a strong contribution of a phonon drag component, we aim for thermal separation between the electron and phonon subsystems and

cool the device to a lattice temperature of 1.33 K. At this temperature, electrons primarily transfer heat to the lattice by emitting acoustic phonons, leading to a much lower electron-phonon relaxation compared to higher temperatures. Additionally, we utilize all-electronic methods of temperature measurement (via electronic noise)[30] and heating (using a heater with hot electron diffusion)[31,32].

As expected for a Weyl semimetal, we observe that the thermal conductance increases with the applied in-plane magnetic field. When we compare the thermal and electrical conductance, we find them matching according to the fundamental Wiedemann–Franz law for all relevant Fermi level positions. This indicates that the gravitational anomaly does not manifest in the heat transport in this system. We discuss the implications of our result for the observation of quantum anomalies in Weyl semimetals.

## Results

For our experiment, we utilize a device shaped into a symmetric H-bar structure[33] (see inset of Fig. 2a). The two 40 μm long and 4 μm wide channels are used as heater and detector. In the center, the channels are connected by a 4.5 μm wide and 6 μm long island of which the transport properties are probed. Both channels and the island are equipped with separate electrostatic gates, allowing for independent control of the carrier density. Unless specified otherwise, we gate the heater and the detector channels into the conduction band, while the gate voltage at the island $V_{g,i}$ is varied to study transport at different positions of the Fermi level.

Figure 2a shows the zero-field longitudinal resistance, $R_{xx}(B = 0)$ of the island at lattice temperature $T_0 \approx 1.33$ K, as the Fermi level is tuned from the conduction band to the valence band. Weyl points are located near the resistance maximum of the gate sweep, corresponding to the charge neutrality point in the band structure. We characterize the island resistance $R_{xx}(B)$ by measuring it as a function of in-plane magnetic field $B$ (applied parallel to the direction of heat transport in the island) at different island gate voltages $V_{g,i}$ using lock-in measurement techniques in Fig. 2b. As shown in Fig. 2b, we observe a negative magnetoresistance below 5 T[16] for a wide range of $V_{g,i}$. The maximum effect is observed around $V_{g,i} = -0.29$ V, where

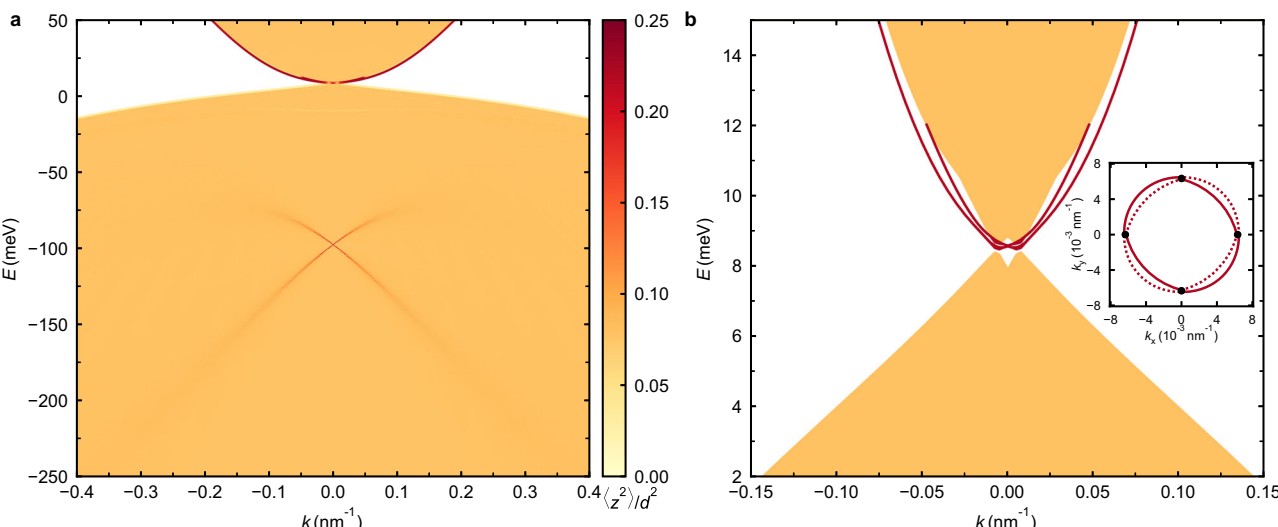

**Fig. 1 | Band structure of compressively strained HgTe. a** Energy momentum dispersion of HgTe under a compressive strain of 0.28%, calculated using $\mathbf{k} \cdot \mathbf{p}$ theory, with momentum $k$ along the (100) direction. In order to analyze the surface states, we use a finite-slab geometry, with surfaces perpendicular to the growth direction (001). The color scale represents the expectation value of the squared coordinate, $\langle z^2 \rangle = \int \Psi(z)z^2\Psi(z)dz$, normalized by the layer thickness $d$, with bulk states shown in orange and surface states in maroon. The massless $n$-type surface states are located near the Weyl points, with the surface Dirac cone situated deep

within the valence band. **b** Magnification of (**a**) near the Fermi level, showing the two Weyl points on the $k_x$ axis with equal chirality and the four $n$-type surface states (maroon). The surface states are two-fold degenerate on the $k_x$ axis, making up for a total of four surface states, i.e., two at each surface. The inner pair merges into the conduction band for $k \approx 0.05$ nm$^{-1}$. The inset depicts the Fermi surface at the Fermi level for the top (solid) and bottom (dashed) surface states. These surface states at the Fermi level are known as "Fermi arcs". The black dots mark the positions of the Weyl points.

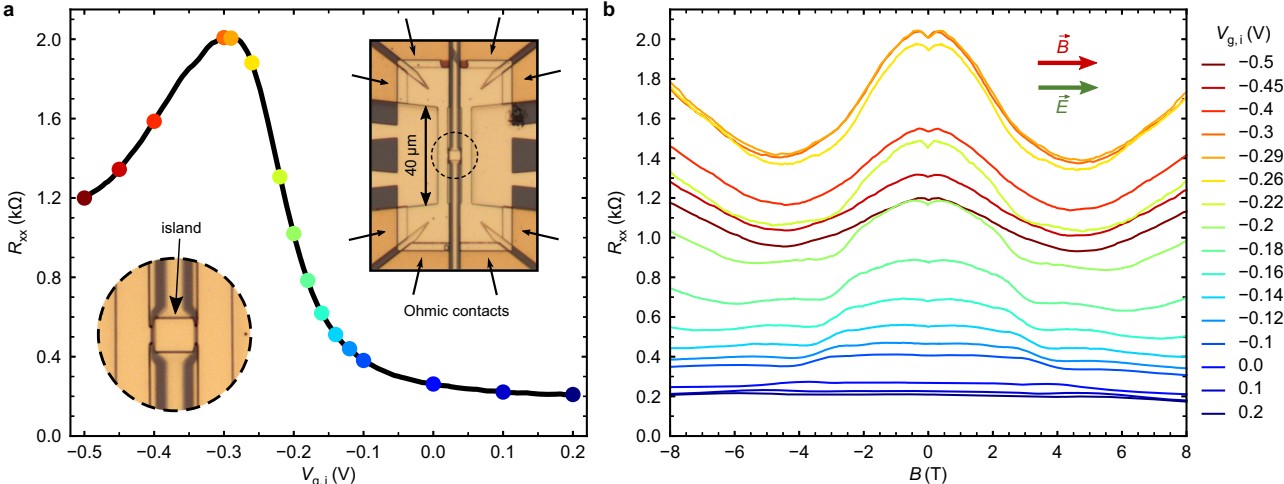

**Fig. 2 | In-plane magnetoresistance of the microstructure fabricated out of a compressively strained HgTe layer. a** Longitudinal resistance $R_{xx}$ as a function of gate voltage $V_{g,i}$ for a $4.5 \times 6$ μm compressively strained HgTe island at zero

magnetic field $B = 0$. Insets: optical micrograph of the full device and a magnified view of the island. **b** In-plane magnetoresistance measured at different Fermi level positions, corresponding to the colored dots in (**a**).

$R_{xx}(0) \approx 2$ kΩ also peaks as a function of $V_{g,i}$ (see Fig. 2a). The negative magnetoresistance occurs exclusively in this field orientation; in all other orientations, we observe a positive magnetoresistance—strong when the magnetic field is perpendicular to the sample plane, and weak when the field lies in-plane but is oriented perpendicular to the current direction (see ref. 16 and Supplementary Note 1).

As discussed extensively in ref. 16, the observed negative magnetoresistance is attributed to the linear band crossings around the Weyl points (i.e., the chiral anomaly), while the small resistance increase up to 0.5 T is related to weak antilocalization.

To study the thermal transport in our Weyl material, we use current heating, running a DC current $I_{heat}$ through the heater channel, which increases the electron temperature at its center to $T_{hot}$, above the lattice temperature $T_0$. This temperature increase results in a heat flow, with rate $Q$, through the island into the detector channel. This heat flow subsequently raises the electron temperature at the detector side, $T_{cold}$.

To determine $T_{cold}$ (and $T_{hot}$), we measure the power spectral density of the voltage fluctuations $S_V$ of the detector channel using a conventional technique with resonant coupling to a cryogenic amplifier[30]. By carefully designing the electrical circuit we ensure that no other parts of the sample contribute to $S_V$, also excluding the potential influence of non-equilibrium noise due to the temperature difference[34,35]. This allows the extraction of an average electronic temperature of the detector $T_{det}$ via the Johnson-Nyquist noise expression $S_V = 4k_B T_{det} R_{ch}$[36], where $R_{ch}$ is the resistance of the channel (Fig. 3a). Using the symmetry of the H-bar, we can obtain the average heater temperature $T_{heat}$ without needing to install and calibrate a second cryogenic amplifier by changing the configuration and passing the current through the detector channel instead (Fig. 3a), while ensuring that $R_{ch}$ is the same for heater and detector channels.

Assuming linear response, the electron thermal conductance $K$ increases linearly with temperature as $K = \kappa T$, where $\kappa$ is the temperature-independent thermal conductance coefficient. The heat transport equation through the island can then be written as (this relation is valid for both coherent and incoherent conductors)[37]

$$Q = \kappa/2(T_{hot}^2 - T_{cold}^2), \qquad (1)$$

where $T_{hot}$ and $T_{cold}$ are the electron temperatures at the entrance and exit of the island, respectively. Since the dominant mechanism of heat loss in the channels is simply through electron diffusion to the leads,

the temperatures inside the channels are not uniform but rather develop profiles towards the leads. $T_{hot}$ and $T_{cold}$ are thus not identical to the measured average temperatures $T_{heat}$ and $T_{det}$, respectively. However, they can be accurately determined from these measurements by taking the temperature profile into account. Meanwhile, the heat flow $Q$ is inferred from the rise in $T_{det}$ and is calibrated using the Joule power required to bring the heater to the same temperature (see Methods for details on both procedures).

Guided by Eq. (1), we plot the measured values of $Q$ against $T_{hot}^2 - T_{cold}^2$ (Fig. 3b). The observed linear relationship confirms that the thermal conductance scales linearly with temperature. For further analysis, we focus on the thermal conductance coefficient $\kappa = K/T$.

In Fig. 3c, the extracted $\kappa$ is shown for different positions of the island Fermi level from $n$-type to $p$-type through the Weyl point. We observe the largest $\kappa$ in the $n$-region (for gate voltage $V_{g,i} = 0.2$ V), where it exhibits a slight monotonous increase with magnetic field. Closer to the Weyl point ($V_{g,i} = -0.2$ V and $V_{g,i} = -0.29$ V), $\kappa$ is more than 6 times smaller and non-monotonous, following $1/R_{xx}$ in its behavior. Perhaps surprisingly, when moving into the $p$-region ($V_{g,i} = -0.4$ V and $V_{g,i} = -0.5$ V) the thermal conductance looks nearly the same as at the Weyl point. We will argue below, that this is simply due to the enhanced hole-phonon scattering due to the large effective mass of the holes in the valence band of HgTe.

In a non-interacting fermion system, charge and heat transport are interconnected, as described by the fundamental Wiedemann-Franz law, $\kappa T = GL_0 T$[13]. This law relates the electrical and thermal conductance through the Lorenz number $L_0$, which for electrons with charge $e$ is given by $L_0 = (\pi^2/3)(k_B/e)^2 \approx 2.44 \times 10^{-8}$ W Ω K$^{-2}$. To check if the Wiedemann-Franz law is obeyed for our Weyl semimetal, in Fig. 3d we show the Lorenz ratio $L = \kappa/G$ with $G = 1/R_{xx}$ taken from the corresponding traces in Fig. 2b. For any $V_{g,i}$ we observe that the variations in magneto-thermal conductance match the variations in magneto-electrical conductance, leading to an almost constant $L$ across the entire studied field range. Importantly, $L \approx L_0$ both at the Weyl point and in the $n$-conducting region. The slight increase of $L$ above $L_0$ at $V_{g,i} = 0.2$ V is well understood since at this gate voltage the measured island resistance is overestimated due to the comparable sheet resistivities of island and channels.

At the same time, we observe $L$ to be significantly lower than $L_0$ in the $p$-regime at $V_{g,i} = -0.4$ V and $V_{g,i} = -0.5$ V. We show in Supplementary Note 4, that in the $p$-regime the heat loss to the phonons in the

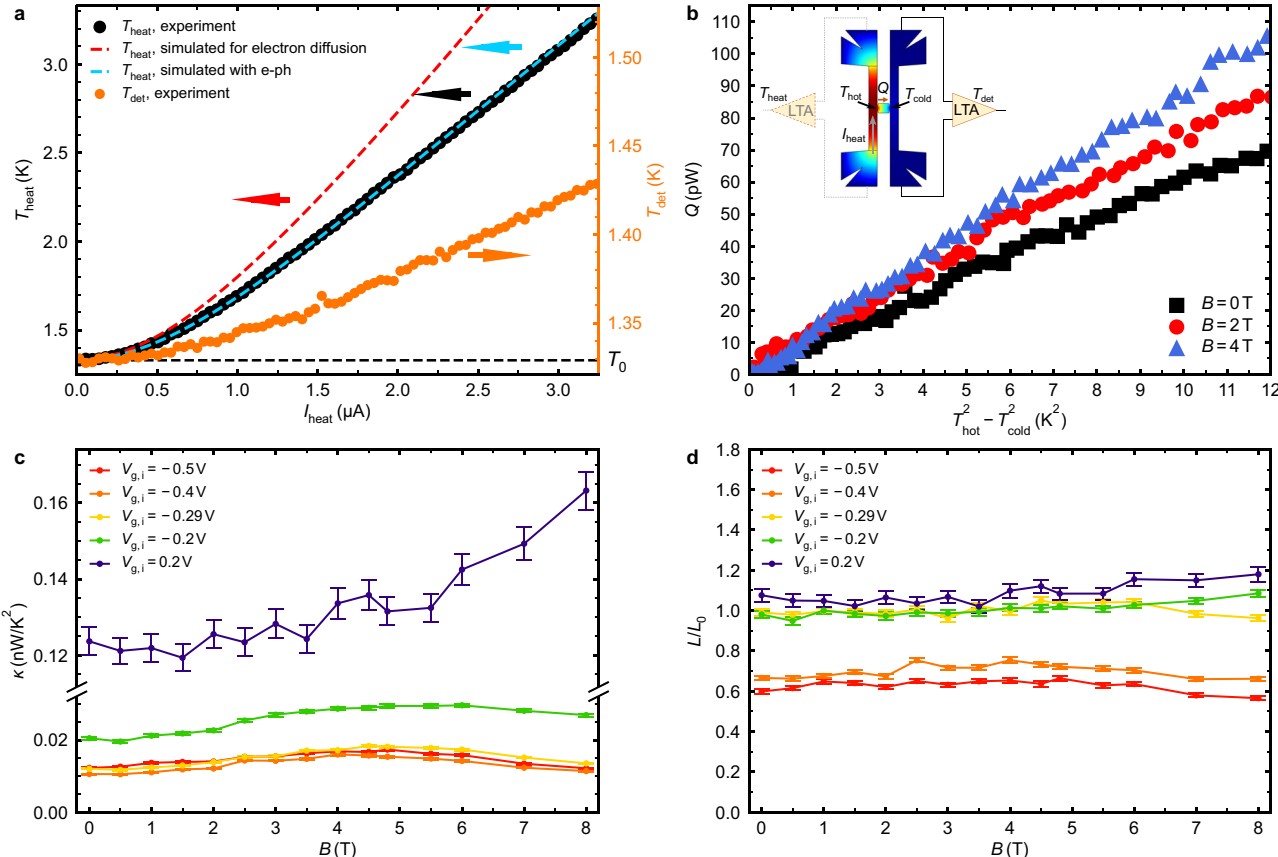

**Fig. 3 | Thermal conductance of the compressively strained HgTe island. a** The average heater channel ($T_{\text{heat}}$, black dots, left axis) and detector channel ($T_{\text{det}}$, orange dots, right axis) temperatures, measured via noise thermometry, as a function of DC heating current, $I_{\text{heat}}$ for in-plane magnetic field $B = 0$. The Fermi level of the island is tuned close to the Weyl point at gate voltage $V_{\text{g,i}} = -0.29$ V. The dashed lines are simulations of the heater performance for pure hot electron diffusion (red) and including electron-acoustic phonon relaxation ($q_{\text{ph}} = \Sigma_{\text{ph}}(T^3 - T_0^3)$ with coefficient $\Sigma_{\text{ph}} \approx 0.19$ Wm$^{-2}$K$^{-3}$) (light blue). Black dashed line marks the base temperature $T_0$ for both axes. **b** The heat flow rate $Q$ through the island as a function of $T_{\text{hot}}^2 - T_{\text{cold}}^2$ for different in-plane magnetic fields: $B = 0$ T, 2 T, and 4 T. The inset shows a schematic of

the thermal transport measurement configuration with the temperature profiles in the channels; LTA denotes a low-temperature (cryogenic) amplifier. Owing to the symmetry of the device, $T_{\text{heat}}$ is measured via running the current through the detector channel, without a second amplifier. **c** Thermal conductance coefficient $\kappa$ of the island as a function of in-plane magnetic field for different positions of Fermi level, ranging from $n$-type regime, through the Weyl point at gate voltage $V_{\text{g,i}} \approx -0.29$ V, and all the way to the $p$-region. **d** Data from (**c**) and Fig. 2b, combined yield the Lorenz ratio $L = \kappa/G$ ($G$ is the electrical conductance) in units of $L_0 \approx 2.44 \times 10^{-8}$ W $\Omega$ K$^{-2}$. The error bars are derived from the statistical analysis of noise measurements, representing a 95% confidence level (see Supplementary Note 7).

island is almost two times larger than both for $n$-regime and close to the Weyl point. This accounts for extra heat loss and reduces the measured $L$. A strong hole-phonon relaxation is expected for HgTe, given the high effective mass of holes leading to much lower mobility and consequently increased scattering.

According to Onsager's generalized transport equations, the presence of thermopower $S$ modifies the thermal conductance in a circuit with no net electric current so that[13,14,38]

$$\kappa = \kappa_0 - S^2 G. \qquad (2)$$

Here, $\kappa_0$ is the intrinsic thermal conductance associated with short-circuited configuration. We note that the effect described by Eq. (2) does not alter the mechanism of heat transport by individual particles but rather describes the macroscopic effect of the presence of a thermoelectric voltage on heat transport, so that the Wiedemann-Franz law should hold for $\kappa_0$. For a proper analysis of $\kappa$, the contribution $S^2 G$ has to be accounted for.

To address this, we measure the development of thermopower across the island with gate voltage using a second harmonic technique. The sign-changing behavior in Fig. 4a is in good agreement with the Mott relation for low temperatures $S \approx -L_0 e T \frac{1}{G} \frac{dG}{d\varepsilon}\big|_{\varepsilon = \varepsilon_{\text{F}}}$, where the derivative of the experimentally measured conductance with energy $\varepsilon$

is taken at the Fermi level $\varepsilon_{\text{F}}$. The observed deviation in magnitude of the experimental value for $S$ is not surprising, as the exact expression for $S$ requires separately accounting for the contributions from all bands[38].

Figure 4b depicts the variation of thermopower with the in-plane magnetic field $B$ at $V_{\text{g,i}} = -0.29$ V. Despite a slight increase with $B$, the contribution $S^2 G$ remains at least three orders of magnitude smaller than the measured thermal conductance coefficient $\kappa$, which thus does not require extra correction.

## Discussion

Weyl semimetals are of significant interest due to their potential for studying quantum anomalies, which play crucial role in understanding exotic transport phenomena. Although negative magnetoresistance is frequently observed in these systems, it can result from any linear crossings in the band structure close to the Fermi energy and does not necessarily serve as definitive evidence of chiral anomaly[7]. In contrast, thermal conductance measurements may provide a more robust indication of the non-trivial transport regime associated with quantum anomalies. For gravitational anomalies, a violation of the Wiedemann-Franz law is expected, which requires more complex assumptions about the system's transport properties, such as those described by the hydrodynamic theory of Lucas et al.[6] Although the hydrodynamic

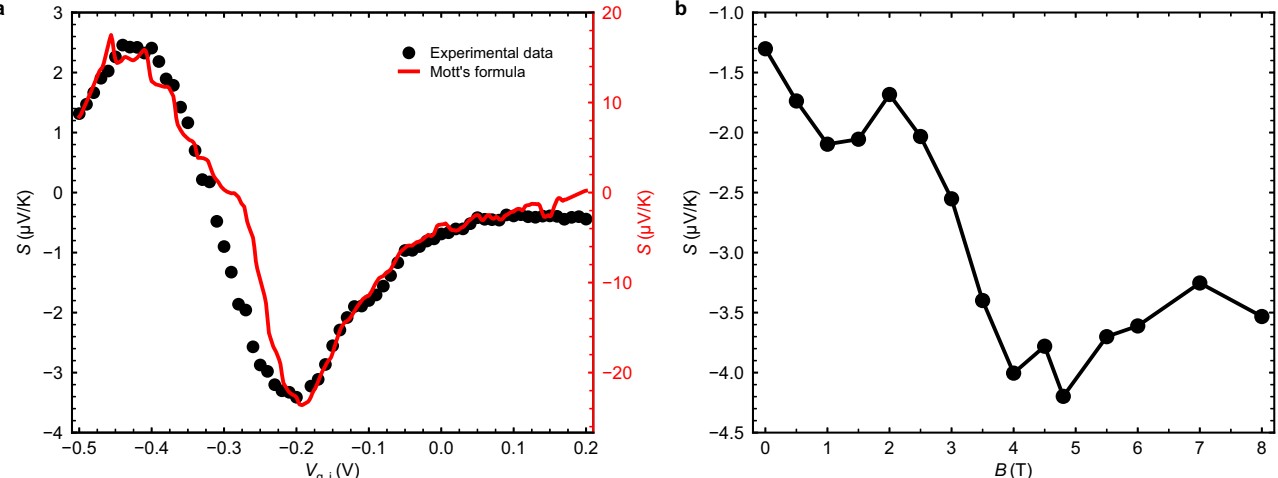

**Fig. 4 | Thermopower of the compressively strained HgTe island. a** The thermopower of the island as a function of the island gate voltage $V_{g,i}$. The Fermi level within the island is adjusted from the $n$-type region (0.2 V) to the $p$-type region (−0.5 V), passing through the Weyl points at $V_{g,i} \approx -0.29$ V. The black dots represent the experimental data, while the red line shows the calculation based on Mott's formula. **b** The thermopower of the island as a function of the in-plane magnetic field when the island is near the Weyl points.

model predicts Wiedemann-Franz behavior ($L = L_0$) for weak internode scattering and at low temperatures, it also predicts a different sign for the Seebeck coefficient $S$, thereby violating the Mott relation and not aligning with our experimental observations.

Andreev and Spivak[8] derived a full set of thermoelectric coefficients of both topological and conventional conductors based on the relevant relaxation times. They predict both Wiedemann–Franz law and Mott relation obeyed for Weyl semimetal at low temperatures, as all transport phenomena in this limit are related to the chiral anomaly. This agrees with our observations, though it does not necessary suggest any distinct physics, as such behavior could simply stem from band crossing effects. However, in the hydrodynamic regime the authors again rather expect both relations to be not valid. Given that the specific transport regimes, particularly the hydrodynamic regime described in refs. 6 and [8], are not realized in our experiment, the observed increase in thermal conductance with magnetic field is likely unrelated to the gravitational anomaly. We now turn to a discussion of how our findings compare to other experimental observations.

A number of experimental studies have been conducted previously to investigate quantum anomalies in thermal transport of Weyl semimetals. A study featuring the Weyl semimetal NbP[10] investigates a concept called "thermoelectric conductance", which is not directly experimentally accessible and can be rather expressed as a product of the Seebeck coefficient and the electrical conductance, $S \cdot G$. Since $S$ is associated with a different coefficient in the constitutive equations than thermal transport, a separate verification of the Mott relation is necessary to identify a non-trivial thermal transport regime. Another study[11] considers the Weyl phase of $Bi_{1-x}Sb_x$, where adherence to the Wiedemann-Franz law has been reported, similar to our results. We note that both experiments[10,11] have been performed without experimental control of the Fermi level position and in a regime with strongly coupled electron and phonon subsystems using lattice heating at high temperatures. The validity of the Wiedemann-Franz law in this limit for the electronic component of thermal conductance is trivial and does not indicate any protected thermal transport, unlike proposed in ref. 11, since strongly coupled electron and phonon subsystems will reach the same local temperature and the heat flows between them are balanced[39].

The Wiedemann–Franz law is generally robust as it is not based on any major approximations other than the description of transport in terms of fermionic quasiparticles. For this law to be violated (other than by the effect of Eq. (2)), a strong electron-electron interaction is required. While increasing the temperature might seem like a viable option to intensify electron-electron scattering, this approach is very limited. As temperature rises, electron-phonon relaxation also becomes more prominent, narrowing the viable temperature range for thermal transport experiments. This effect can be partially mitigated in materials where the electron-phonon relaxation is suppressed—such as in graphene, due to its high optical phonon energy and weak coupling to acoustic phonons[40]. There, deviations from the Wiedemann-Franz law at zero magnetic field, attributed to hydrodynamic carrier flow, have been observed[41]. However, the same conditions do not apply to Weyl materials, raising doubts about the feasibility of detecting transport anomalies at high temperatures. In contrast, our experiment demonstrates that at low temperatures the anticipated special mechanisms of heat transport are not active.

We conclude that in a Weyl semimetal based on a compressively strained HgTe layer, we observe an increase of electrical and thermal conductances with in-plane magnetic field matching the direction of electric field or temperature difference, respectively. However, these conductances are still perfectly matched by the Wiedemann-Franz law, and the Seebeck coefficient remains in accordance with the Mott relation. These observations are in stark contrast with theories involving the effects of gravitational anomaly, and rather point towards a weak interaction between electrons at different Weyl nodes. The fact that our experiment is performed at low temperature, where little phonon scattering occurs, ensures that the intrinsic carrier behavior is not influenced by phonon drag, which is otherwise a very common phenomenon in semiconductor materials. At the same time, we gathered a complete set of transport coefficients for thermoelectric transport in a Weyl semimetal, which can serve as a basis for development of a transport theory in such materials.

## Methods
### Material and fabrication
The device is fabricated on a 72 nm thick HgTe layer sandwiched between two 15.2 nm top and bottom (Hg,Cd)Te layers on a GaAs substrate, grown by MBE. A CdTe/ZnTe virtual substrate applies a 0.28% compressive strain, lifting the $\Gamma_8$ band degeneracy at the $\Gamma$ point of the Brillouin zone to form Weyl points. The presence of the 0.28% compressive strain was confirmed by high-resolution X-ray diffraction (HRXRD) measurements, as discussed in detail in ref. 16.

The H-bar device consists of two channels connected by a central island. The channels are 40 μm long and 4 μm wide, while the island

measures 6 μm in length and 4.5 μm in width. The device is patterned using electron beam and optical lithography in conjunction with wet etching. Ohmic contacts to the device are fabricated by depositing AuGe and Au layers, each with a thickness of 80 nm. The top gate electrodes are composed of 5 nm Ti and 100 nm Au, with a 15 nm HfO$_2$ gate insulator, grown via atomic layer deposition, separating them from the mesa. The wide Ohmic contacts, fabricated using conventional lithography techniques, ensure homogeneous current injection and thereby eliminate current-jetting effects[42], while the electrostatic gate enables tuning of the Fermi level in the thin layer to the Weyl point.

The carrier density and mobility were determined from Hall measurements performed on a gated Hall bar device (dimensions: 1200 μm × 200 μm) fabricated from the same wafer (Supplementary Note 8). Within the applied gate voltage range, the carrier density can be tuned from $n \approx 3.2 \times 10^{11}$ cm$^{-2}$ at $V_{g,i} = 0.1$ V to $n \approx -2.2 \times 10^{11}$ cm$^{-2}$ at $V_{g,i} = -0.5$ V. According to our band structure calculations, the Fermi level remains within a few meV of the Weyl points throughout this range. From the zero-field conductance, we estimate the mobility in the $n$-type conduction ($V_{g,i} = 0.1$ V) regime to be $\mu \approx 60 \times 10^3$ cm$^2$/(Vs). The mobility decreases as the Fermi level approaches the Weyl point, reaching $\mu \approx 22.8 \times 10^3$ cm$^2$/(Vs) at $V_{g,i} = -0.2$ V.

## Transport measurements

The device is cooled in exchange gas thermally connected to a pumped $^4$He cryostat, resulting in base temperature $T_0 \approx 1.33$ K. The copper chip carrier is attached to a copper sample receptacle, resulting in a large effective cooling area and stable phonon temperature. In close proximity of the sample, a calibrated resistance thermometer is attached, allowing to measure the lattice temperature.

The device resistance is measured using a standard lock-in technique, wherein an alternating current with a frequency of ≈3 Hz and an amplitude of less than 60 μV is applied across the device and a 100 kΩ series resistor, across which the current is measured. The resistance of the island is measured using a four-terminal configuration, with the legs of the H-bar serving as the current and voltage probes. Magnetic field dependent properties are measured by applying a magnetic field at a sweeping rate of 0.1 T/min, aligned with the direction of heat and charge flow in the island and within the device plane.

## Noise thermometry

We measure the power spectral density $S_V$ from the voltage fluctuations picked up from the detector channel using a home-built low-temperature amplifier (LTA) with a voltage gain $A_{LT} \approx 5$ (the gain is determined by calibration of the measurement circuit using its thermal noise, the long-term stability is $\delta A_{LT}/A_{LT} < 1\%$), connected to one side of the detector channel through a coaxial cable. The amplifier is located outside of the sample chamber, submerged in a pumped liquid helium bath, away from the magnetic field. An air-core superconducting inductor is connected in parallel to the amplifier input, resulting in combination with the coaxial cable capacitance in resonance impedance matching at $f_{res} \approx 5.7$ MHz. The other side of the detector channel is effectively grounded at $f_{res}$ using a 10 nF capacitor. The remaining lines are choked for high frequency signals using 50 kΩ resistors, still allowing to apply and measure DC signals (see the full measurement schematic in Supplementary Note 2). This circuit results in a minimal pickup of the electronic noise from the heater side and the island itself, effectively measuring the noise temperature of the detector only.

The amplified voltage noise is guided outside of the low-temperature compartment to a room temperature amplifier with a voltage gain $A_{RT} \approx 200$ and finally into a spectrum analyzer. The spectrum analyzer measures the average square voltage $V_{SA} = \sqrt{\langle V^2 \rangle}$

around $f_{res}$ in a 30 kHz bandwidth ($\Delta f$) defined with a Gaussian filter. We convert $V_{SA}$ to spectral power density of the voltage fluctuations $S_V$ using $S_V = V_{SA}^2/(\Delta f A_{LT} A_{RT})$.

When the heating current is applied to the side opposite the LTA connection, the electronic noise is converted to the average detector temperature using the Johnson-Nyquist relation $S_V = 4k_B T_{det} R_{ch}$[36]. Similarly, when the heating current is applied on the same side as the LTA connection, we extract the average heater temperature using: $S_V = 4k_B T_{heat} R_{ch}$. In this latter case, the extraction is reliable, as the resistance of the heater channel remains almost constant with heating current (see Supplementary Note 3).

## The heat transport regime of heater and detector

Accurate extraction of the thermal conductance requires a clear understanding of the heat transport regime in the heater and detector channels. We identify this regime by analyzing the average heater temperature $T_{heat}$ as a function of the heating current $I_{heat}$ in a broad range, extending up to $T_{heat} \approx 12$ K. The observed behavior reflects the underlying electron diffusion and electron-phonon relaxation mechanisms[43–46].

Without allowing for phonon emission by hot electrons, one expects $T_{heat} = T_0/2 [1 + (\nu + \nu^{-1}) \arctan(\nu)]$[45] with the dimensionless parameter $\nu = \sqrt{3} e V_{bias}/(2\pi k_B T_0)$, where $V_{bias} = I_{heat} R_{ch}$ is the voltage bias on the channel. This formula describes a transition from quadratic heating with $T_{heat} \approx T_0(1 + \nu^2/2)$ at $eV_{bias} \ll k_B T_0$ to a linear temperature increase with $T \approx (\sqrt{3}/4) eV_{bias}/(2k_B)$ when $eV_{bias} \gg k_B T_0$. In the range up to $T_{heat} \sim 3$ K, we rather observe a slower growth of $T_{heat}$ (Fig. 3a). For higher temperatures, we observe a characteristic bending (Supplementary Note 4), indicating a phonon contribution to the heat relaxation in channels.

We identify the phonon relaxation law by matching the experiment with a finite element simulation based on the heat transport equation (see Supplementary Note 4). We find the electron-phonon relaxation per area in a form $q_{ph} = \Sigma_{ph}(T^3 - T_0^3)$ with coefficient $\Sigma_{ph} \approx 0.19$ Wm$^{-2}$K$^{-3}$. The power 3 corresponds to the relaxation via emission of polar acoustic phonons at low temperature[47]. A relaxation rate following the same dependence has been observed previously for narrow HgTe quantum wells[48].

## Determination of local temperatures $T_{hot}$ and $T_{cold}$

Johnson–Nyquist thermometry provides a measure of the average electron temperatures $T_{heat}$ and $T_{det}$. However, Eq. (1) involves local temperatures, which differ due to spatial temperature profiles along the channels.

Assuming electronic diffusion is the dominant heat transport mechanism, the heat equation can be solved analytically for a uniform strip. This yields

$$T_{heat} = \frac{1}{2} \left[ T_0 + \frac{T_{hot}^2}{\sqrt{T_{hot}^2 - T_0^2}} \arccos \frac{T_0}{T_{hot}} \right],$$

$$T_{det} = \frac{2}{3} \left[ T_0 + T_{cold} - \frac{T_{cold} T_0}{T_{cold} + T_0} \right]. \tag{3}$$

The difference between these expressions arises from the distinct heating geometries: in the heater channel heat is generated uniformly along its length, whereas in the detector channel it is injected locally at the center.

The analysis of heat transport in the channels reveals that electron-phonon relaxation plays a significant role, thereby modifying the spatial temperature profiles and potentially affecting the result of Eq. (3). To account for this, we determine the actual local temperatures $T_{hot}$ and $T_{cold}$ through detailed numerical modeling, which incorporates both electron-phonon coupling and the full

device geometry (see Supplementary Note 6). We find that in the heater channel, the relation $T_{hot}(T_{heat})$ still closely follows the prediction of Eq. (3), while in the detector channel, the local temperature satisfies $T_{cold}/T_0 - 1 \approx 2.63 \cdot (T_{det}/T_0 - 1)$ within the relevant range.

## Evaluation of the heat flow rate $Q$

The average temperature increase of the detector channel, $T_{det}$ reflects the heat flow $Q$ entering the detector through the island. To determine $Q(T')$ for a given detector temperature $T_{det} = T'$, we identify the heater current $I'_{heat}$ that brings the heater channel to the same average temperature, such that $T_{heat}(I'_{heat}) = T'$. The corresponding Joule power in the heater, $P_J = (I'_{heat})^2 R_{ch}$, serves as a reference for calibrating the detector response, as the channels are identical in their thermal transport properties (see Supplementary Note 5). However, $Q$ and $P_J$ are not identical, as the power required to reach a given temperature depends on whether heat is introduced uniformly or locally within the channel.

In the limit of small temperature increase above the base temperature $T_0$, and assuming purely electronic diffusive heat transport without phonon losses for a uniform strip, the average temperatures are given by[49]

$$
\begin{aligned}
T_{heat} &= T_0 + (P_J R_{ch})/(12 L_0 T_0), \\
T_{det} &= T_0 + (Q R_{ch})/(8 L_0 T_0).
\end{aligned}
\tag{4}
$$

The difference between these expressions stems from the fact that the heater is uniformly heated by Joule dissipation, while the detector experiences localized heating at its center due to the incoming heat flow. Matching $T_{heat} = T_{det} = T'$ yields $Q = \alpha_Q P_J$ with $\alpha_Q = 2/3$. The prefactor accounts for different spatial profiles of heating. For greater accuracy, we refine $\alpha_Q$ using a numerical model that incorporates the actual device geometry and electron-phonon relaxation. The resulting calibration factor $\alpha_Q$ lies in the range $0.613 \le \alpha_Q \le 0.625$, with weak temperature dependence (see Supplementary Note 6).

Using the above, our calibration procedure is:
1. We measure $T_{heat}$ as a function of the heater current.
2. To find the heat flow for some detector temperature $T' = T_{det}$ we interpolate $T_{heat}$ to find $I'_{heat}$ where $T_{heat}(I'_{heat}) = T'$.
3. We calculate the heat flow $Q(T') = \alpha_Q (I'_{heat})^2 R_{ch}$.

This method provides a robust and self-calibrated value of $Q$, where the influence of device geometry and phonon losses are accounted for through $\alpha_Q$. As a result, uncertainties in these effects contribute only as second-order corrections to the final value of $Q$.

## Thermovoltage measurement

For a thermovoltage measurement, we pass a low-frequency alternating current through one of the channels, thereby inducing a time-dependent electron temperature difference across the island. The resulting thermal voltage develops at twice the excitation frequency and is measured using a standard lock-in technique at the second harmonic frequency. This thermal voltage is subsequently converted to the Seebeck coefficient $S$, utilizing the electron temperature difference between the heater and detector channels, which is determined via Johnson noise thermometry in the regime where the temperature has quadratic dependence on the heating current (close to $T_0$).

## Data availability

The data supporting the findings of this study are available at https://doi.org/10.5281/zenodo.15530401.

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

## Acknowledgements
We thank Mohamed Abdelghany, Yuan Yan, and Mitali Banerjee for their help in developing the Johnson-Nyquist thermometry technique, Piotr Surówka, Roderich Moessner, and Francisco Peña-Benitez for useful discussions on electron heat transport in Weyl semimetals, and Andrew Mackenzie and Claudia Felser for hosting our Max-Planck Fellowship. This work was supported by the DFG through project SFB 1170 (Project ID 258499086) and the Würzburg-Dresden Cluster of Excellence on Complexity and Topology in Quantum Matter (EXC 2147, Project ID 390858490) [J.K., H.B., L.W.M.]; by the Free State of Bavaria through the Institute for Topological Insulators; and through a Max Planck fellowship at the Max Planck Institute for Chemical Physics of Solids, Dresden [L.W.M.].

## Author contributions
A.A.A., S.U.P., H.B., and L.W.M. planned and designed the experiment. D.C. grew the material, and F.S. fabricated the HgTe device under supervision of J.K. A.A.A. performed the experiments and carried out the numerical simulations assisted by S.U.P. and W.B. Y.J.H. contributed in developing the measurement setup for noise thermometry. All authors participated in the analysis led by A.A.A. and S.U.P. All authors participated in writing of the manuscript.

## Funding

## Competing interests
The authors declare no competing interests.
