## [Transparent Peer Review file · Nature Communications]

Wiedemann–Franz behavior at the Weyl points in compressively strained HgTe

Corresponding Author: Dr Stanislau Piatrusha

Version 0:

Reviewer comments:

Reviewer #1

(Remarks to the Author)

Dear Editor,

After carefully reviewing this manuscript, I am impressed by the innovative and rigorous measurement techniques employed. The experimental data sufficiently support the core conclusion that "the low-temperature thermal transport of HgTe under stress still conforms to the Wiedemann-Franz law." However, the authors further assert that the compressively strained HgTe transforms into a Weyl fermion system, and that thermal transport does not violate the Wiedemann-Franz law. This claim lacks sufficient justification, and there are several specific issues that need clarification from the authors:

1. Mechanism of Negative Magnetoresistance

As stated by the authors, negative magnetoresistance may arise from multiple mechanisms, and the positive magnetoresistance observed when the magnetic field exceeds 2T does conform to weak localization characteristics. However, the key question is: how do the authors rule out that the negative magnetoresistance in the magnetic field region below 2T is not caused by weak anti-localization effects? This requires more substantial experimental evidence for clarification.

2. Completeness of Gate Voltage Regulation

The study achieves continuous control of the Fermi surface between the valence band and the conduction band through gate voltage. However, the data presented indicate that when $V_g = -0.29V$, the Fermi surface should be precisely located at the Weyl point, at which point a significant positive magnetoresistance effect should be observed under vertical magnetic field conditions. The absence of this critical experimental data weakens the conclusion regarding Weyl semimetals.

3. Contradictory Phenomena in Magnetic Field Response

An analysis combining Figures 2b and 3d reveals that although a 2T magnetic field is a turning point for the dominance of chiral anomalies and weak localization, the Lorenz constant shows an unusually smooth variation within the range of 0-8T. Does this suggest that the thermal response across the entire magnetic field range (both $<2T$ and $>2T$) originates from trivial electronic states rather than Weyl fermions?

Considering the innovative thermal measurement techniques in this work and the cutting-edge nature of the research topic, I recommend that it is suitable for publication in Nature Communications after the above key evidence is supplemented, particularly providing conclusive experimental evidence that the material system possesses Weyl semimetal characteristics.

Reviewer #2

(Remarks to the Author)

This paper by A. Aravindnath, et al investigates the Wiedemann-Franz behavior in a Weyl semimetal based on compressively strained HgTe. By performing thermometry measurements at low temperatures using fully electronic methods,

the research clearly identifies the Weyl regime and extracts the thermal conductance. The observed increase in thermal conductance with the applied magnetic field matches the increase in electrical conductance, adhering to the Wiedemann-Franz law. This indicates that, despite the unique electronic structure of Weyl semimetals, the thermal transport mechanism remains consistent with electrical transport, without violating any conservation laws. Overall, this work is interesting and novel, and I would be willing to recommend it for publication in Nature Communications after the authors address the following concerns:

As we know, in thermal conductance measurements, the heat flow or heat power Q flowing through the sample is a crucial and intuitive quantity, especially when studying scaling laws such as the Wiedemann-Franz law. However, the analysis and calculation of the heat power Q in the manuscript appear overly simplistic and cursory. I suggest that the authors provide a clearer and more detailed explanation of this heating power.

Reviewer #3

(Remarks to the Author)

This paper mainly studies the thermal transport properties of compressively strained HgTe layer. The Fermi level can be tuned by changing the gate voltage. The authors found that even though Weyl fermions are located at the Fermi level and dominate transport properties, the Wiedemann-Franz law still holds, indicating that Weyl fermions do not cause distinct physics in thermal transport. In my opinion, this paper does not meet the high standards of Nature Communications, and other journals such as Communications Physics may be more suitable.

(1) The innovation of the paper is not sufficient to be published in NC. I don't think the results of this article are very interesting.

(2) In the "results" section, the author spent a considerable amount of space describing how the experiment was conducted, which should be included in the "methods" section. The main text should mainly describe and discuss the experimental results.

(3) The data is not abundant and the quality is not high.

Version 1:

Reviewer comments:

Reviewer #1

(Remarks to the Author)

This article elaborates on the research conducted to modulate the topology of HgTe by precisely applying external stress to alter its crystal structure. This work represents significant advancement and importance in the field of topological materials. However, the determination of whether the modulation is successful must be approached rigorously, primarily based on the following two criteria: 1) Whether the crystal structure has indeed changed. This serves as the foundation for verifying the modulation effect. 2) Whether the key experimental evidence supporting the two types of topological properties is sufficient. This is crucial for ensuring the reliability of the research conclusions.

As these critical conditions have not yet been fully verified, it is currently not possible to scientifically conclude whether this research has achieved its intended goals.

Reviewer #2

(Remarks to the Author)

All my concerns have been addressed. I recommend its publications.

Reviewer #3

(Remarks to the Author)

Version 2:

Reviewer comments:

Reviewer #1

(Remarks to the Author)

Dear Editor,

I have carefully read the authors' responses and can see that they have addressed my questions with serious consideration. In principle, I agree to accept the publication of this manuscript, mainly for two reasons: (1) the authors' methodology and overall approach are highly advanced, and their experimental techniques show significant innovation; and (2) at present, there is no universally accepted or entirely conclusive experimental evidence for transport signatures in topological semimetals. While some theoretical papers have proposed that negative magnetoresistance and the planar Hall effect could serve as evidence of topological semimetals, in practice, when tested in real materials, almost none have fully satisfied

these criteria. This work represents an important exploration in this area and is likely to attract further attention from theorists and researchers with relevant expertise, stimulating additional investigation into this timely scientific problem. For these reasons, I support its publication.

Best
Chen

RESPONSE TO REVIEWERS' COMMENTS

Reviewer #1 (Remarks to the Author):

Dear Editor,

After carefully reviewing this manuscript, I am impressed by the innovative and rigorous measurement techniques employed. The experimental data sufficiently support the core conclusion that "the low-temperature thermal transport of HgTe under stress still conforms to the Wiedemann-Franz law." However, the authors further assert that the compressively strained HgTe transforms into a Weyl fermion system, and that thermal transport does not violate the Wiedemann-Franz law. This claim lacks sufficient justification, and there are several specific issues that need clarification from the authors:

The authors' reply:

We thank the reviewer for carefully evaluating the manuscript and appreciate the constructive criticism. Please find our detailed response below.

1. Mechanism of Negative Magnetoresistance

As stated by the authors, negative magnetoresistance may arise from multiple mechanisms, and the positive magnetoresistance observed when the magnetic field exceeds 2T does conform to weak localization characteristics. However, the key question is: how do the authors rule out that the negative magnetoresistance in the magnetic field region below 2T is not caused by weak anti-localization effects? This requires more substantial experimental evidence for clarification.

The authors' reply:

Compressively strained HgTe exhibits a pronounced negative magnetoresistance (MR) when the magnetic and electric fields are aligned (i.e., $B \parallel E$). This negative magnetoresistance, however, is accompanied by positive MR features at both low and high magnetic field ranges. The positive magnetoresistance at low fields ($< 0.5T$) is indeed attributed to the weak anti-localization (WAL) effect, a quantum interference phenomenon. The reviewer is probably referencing the weak localization (WL), another quantum interference phenomenon, that leads to resistance decrease with magnetic field and is commonly observed in combination with WAL.

Several arguments allow us to clearly distinguish the observed resistance decrease at intermediate fields (from 0.5T up to 5T) from WL:

First, due to the high mobility in our sample, the observed negative MR cannot be attributed to WL. Instead, at the expected very low magnetic fields, HgTe exhibits WAL (Ref. A), arising from strong spin-orbit coupling, on top of the negative MR background.

Second, WL is expected to be active when electrons can pick phase from magnetic fields. This is typically stronger for the field orientations, perpendicular to the current flow, due to the larger flux through electron trajectories. In contrast, in our system negative MR is only observed for in-plane magnetic field, parallel to the current flow, $B \parallel I$. For all other field directions, we observe positive MR (see the answer to the next question and newly added Supplementary Fig. 1 in the Supplementary

Information). Specifically, for in-plane perpendicular we observe very weak positive MR, which is unlikely for WL.

Third, the observed negative MR in compressed HgTe survives at high temperatures, while WL is suppressed quickly as the quantum coherence is destroyed. Below we present the data adopted from the thesis of David Mahler (<https://nbn-resolving.org/urn:nbn:de:bvb:20-opus-253982>, Fig.7.8), where the temperature dependence of magnetoresistance in compressively strained HgTe was measured under $B \parallel E$. The measurements were performed in our group on a different device with the layer stack and material properties closely matching those of the device discussed in this work. On top of the original figure panel b we added a mark pointing to the region of WAL.

[Figure Redacted]

From Fig.7.8 (b), we observe that the sharp WAL feature at $B=0T$ quickly disappears with increasing temperature, confirming its origin from interference effects. In contrast, the pronounced negative MR at intermediate magnetic fields remains largely unchanged. This effectively rules out quantum interference mechanisms such as WL as the underlying cause. A noticeable change in the negative magnetoresistance is only observed at 23 K, likely due to enhanced phonon scattering at elevated temperatures.

Last, in our device we have a direct control over the position of the Fermi level in the band structure with an electrostatic gate. We observe prominent negative MR only when the Fermi level is tuned near the position of the Weyl points, which is verified via carrier density extracted from the out-of-plane magnetic field Hall measurement. This does not match the expectations for WL, where no direct relation to the band structure position of Fermi level is expected. At the same time, the band structure calculations based on $\mathbf{k} \cdot \mathbf{p}$ theory reveal the existence of linearly dispersing Weyl points in compressively strained HgTe, that should result in effect of chiral anomaly only for Fermi level close the Weyl points as detailed in Ref. [16].

These directional and energy-specific properties of our observations support the interpretation of the negative magnetoresistance as a signature of the chiral anomaly.

2. Completeness of Gate Voltage Regulation

The study achieves continuous control of the Fermi surface between the valence band and the conduction band through gate voltage. However, the data presented indicate that when $V_g = -0.29V$, the Fermi surface should be precisely located at the Weyl point, at which point a significant positive magnetoresistance effect should be observed under vertical magnetic field conditions. The absence of this critical experimental data weakens the conclusion regarding Weyl semimetals.

The authors' reply:

We have added measurements of the magnetoresistance of our device for out-of-plane magnetic field configuration to the Supplementary Materials. Indeed, for an out-of-plane field, when the

Fermi level is tuned to the Weyl points, a pronounced positive magnetoresistance is observed. We also performed measurements with an in-plane magnetic field, perpendicular to the electric field, which results in very weak positive magnetoresistance. These observations indicate that the negative magnetoresistance is exclusive to the configuration where the magnetic field is aligned parallel to the electric field within the device, as expected for chiral anomaly. A comprehensive study on the angular dependence of magnetoresistance actually is available in our previous paper on compressively strained HgTe with a comparable layer stack and electronic properties (Ref. [16]), providing further insights into the role of chiral anomaly in the observed transport behavior.

3. Contradictory Phenomena in Magnetic Field Response

An analysis combining Figures 2b and 3d reveals that although a 2T magnetic field is a turning point for the dominance of chiral anomalies and weak localization, the Lorenz constant shows an unusually smooth variation within the range of 0-8T. Does this suggest that the thermal response across the entire magnetic field range (both <2T and >2T) originates from trivial electronic states rather than Weyl fermions?

The authors' reply:

We thank the reviewer for their useful comment, that points directly to the core message of our study. The stability of the Lorenz ratio as a function of the magnetic fields demonstrates that the heat is transported via the same mechanism as the charge at all fields, and that it is transported via fermionic quasiparticles with charge $-e$. So, this does not mean that the thermal transport is trivial, on the contrary, this indicates, that if the charge transport originates from the effect of the chiral anomaly, so does the heat transport.

However, theories of Weyl semimetals predict that their thermal conductance can be influenced by the so-called gravitational anomaly. Below we summarize how our observations relate to expectations across different transport regimes (we only consider those, where negative magnetoresistance from chiral anomaly should be present):

1. In an ultraclean Weyl semimetal, the chiral anomaly might affect the charge transport due to the anomalous charge pumping between the Weyl nodes, but less so the energy and heat transport. In that case one might expect $L < L_0$.
2. If inelastic e-e interactions are strong and dominate over all other elastic and inelastic scattering processes, a hydrodynamic transport regime might be realized, where electrons exhibit a form of collective motion, which should result in a substantial breakdown of the Wiedemann-Franz law, presumably with $L > L_0$. This is the regime where the gravitational anomaly effects might be observed in thermal transport.
3. Otherwise, if inelastic electron-phonon scattering dominates, which is expected for higher temperatures, electrons and phonons are strongly coupled and come to the local equilibrium. Thermal conductance then contains electron and phonon components, and the electronic part trivially follows $L=L_0$.
4. If elastic scattering dominates and the internode scattering rate is low, both charge and heat are transported via the same fermionic quasiparticles, and $L=L_0$ is expected, even in the presence of the anomaly.

Previous studies of thermal transport in Weyl systems were done in regime (3), due to the measurements being performed at high lattice temperatures, when electron-phonon scattering is strong. In contrast, we take special care to ensure that we are not operating in this regime by working at low lattice temperatures with fully electronic methods of heating and temperature detection, minimizing the effects of phonon drag and lattice heating. We also directly verify that the electron-phonon relaxation is small enough. However, we find, that under these conditions $L=L_0$, indicating that regime (2) is not realized and rather pointing to the regime (4), which is by no means trivial, but does not allow to experimentally detect the effect of gravitational anomaly.

Considering the innovative thermal measurement techniques in this work and the cutting-edge nature of the research topic, I recommend that it is suitable for publication in Nature Communications after the above key evidence is supplemented, particularly providing conclusive experimental evidence that the material system possesses Weyl semimetal characteristics.

Reviewer #2 (Remarks to the Author):

This paper by A. Aravindnath, et al investigates the Wiedemann-Franz behavior in a Weyl semimetal based on compressively strained HgTe. By performing thermometry measurements at low temperatures using fully electronic methods, the research clearly identifies the Weyl regime and extracts the thermal conductance. The observed increase in thermal conductance with the applied magnetic field matches the increase in electrical conductance, adhering to the Wiedemann-Franz law. This indicates that, despite the unique electronic structure of Weyl semimetals, the thermal transport mechanism remains consistent with electrical transport, without violating any conservation laws. Overall, this work is interesting and novel, and I would be willing to recommend it for publication in Nature Communications after the authors address the following concerns:

As we know, in thermal conductance measurements, the heat flow or heat power Q flowing through the sample is a crucial and intuitive quantity, especially when studying scaling laws such as the Wiedemann-Franz law. However, the analysis and calculation of the heat power Q in the manuscript appear overly simplistic and cursory. I suggest that the authors provide a clearer and more detailed explanation of this heating power.

The authors' reply:

We thank the reviewer for their useful comments and appreciate their evaluation of our work. In the updated version of the manuscript, we reworked and expanded the discussion of the extraction of the heat flux, flowing through the sample and added extra details to support our analysis. We added extra subsection to the "Methods" that explains our approach to finding the heat flux.

Reviewer #3 (Remarks to the Author):

This paper mainly studies the thermal transport properties of compressively strained HgTe layer. The Fermi level can be tuned by changing the gate voltage. The authors found that even though Weyl fermions are located at the Fermi level and dominate transport properties, the Wiedemann-Franz law still holds, indicating that Weyl fermions do not cause distinct physics in thermal transport. In

my opinion, this paper does not meet the high standards of Nature Communications, and other journals such as Communications Physics may be more suitable.

(1) The innovation of the paper is not sufficient to be published in NC. I don't think the results of this article are very interesting.

The authors' reply:

We respectfully disagree with the reviewer on this point. In our view, the present work makes a meaningful contribution to the study of Weyl semimetals. The question of whether the Wiedemann–Franz law is violated—and whether hydrodynamic electron thermal conductance can be detected—is central to experimental efforts aimed at observing the solid-state analogue of the gravitational anomaly in these systems. In fact, the original proposal that Weyl semimetals could serve as a platform for such studies has been a key motivation for much of the thermal transport research in this field.

However, a reliable experimental confirmation of nontrivial heat transport mechanisms in Weyl semimetals remains elusive. This is due in part to the challenge of reaching a regime where electron–phonon scattering is weak, and in part to the intrinsic difficulty of isolating the electronic contribution to thermal conductance. In this work, we present an experiment that explicitly addresses both issues with care—yet we still do not observe the expected thermal anomaly. We believe this result provides important insight into the nature of thermal transport in this class of materials.

(2) In the “results” section, the author spent a considerable amount of space describing how the experiment was conducted, which should be included in the “methods” section. The main text should mainly describe and discuss the experimental results.

The authors' reply:

We acknowledge that the details of Johnson-Nyquist thermometry might not be interesting for every reader of the manuscript. Yet in our case the choice of the method of thermal conductance measurement is strongly connected to the observed physics: we want to operate in a regime where the phonon contribution is small and electron-phonon coupling is weak enough to be able to resolve the specific mechanisms of electronic transport. In the updated version of the manuscript, we moved some less important description of the measurement into the methods section, while keeping the necessary details to support this message.

(3) The data is not abundant and the quality is not high.

The authors' reply:

We would like to draw the reviewer's attention to the details of our study, in particular, to the research method we used. High precision Johnson-Nyquist thermometry is a very time-consuming experimental technique, requiring prolonged measurements to achieve high precision due to the statistical nature of the signal. The value that one should look at to understand the final precision is the relative error $\Delta T/T$. In our case we achieved ΔT of approx. 5mK for $T \sim 1.3\text{K}$ and 95% confidence interval. By this parameter our measurement is on par with the best practices in the field.

Additionally, to improve this parameter by n times, the measurement time needs to be multiplied by n^2 . This can potentially result in enormous measurement times, that don't necessarily improve the analytic potential of the data. A study that results in high precision thermal transport coefficients, however, still requires several months of continuous measurements.

The choice of the research method is not our whim - the fragility of electronic thermal transport with respect to electron-phonon interactions and the need to accurately control heat flows in the system simply does not allow the use of other approaches, which we discuss in detail in the text of the article.

Given all this, we believe that the quantity of data and their quality are sufficient to reliably support the main conclusions of our article and follow the best practices in the field.

- A) Mühlbauer, M., et al. "One-dimensional weak antilocalization due to the Berry phase in HgTe wires." *Phys. Rev. Lett.* 112, 146803 – 2014.

RESPONSE TO THE REFEREE COMMENTS

Reviewer #1 (Remarks to the Author):

This article elaborates on the research conducted to modulate the topology of HgTe by precisely applying external stress to alter its crystal structure. This work represents significant advancement and importance in the field of topological materials. However, the determination of whether the modulation is successful must be approached rigorously, primarily based on the following two criteria: 1) Whether the crystal structure has indeed changed. This serves as the foundation for verifying the modulation effect. 2) Whether the key experimental evidence supporting the two types of topological properties is sufficient. This is crucial for ensuring the reliability of the research conclusions.

As these critical conditions have not yet been fully verified, it is currently not possible to scientifically conclude whether this research has achieved its intended goals.

The authors' reply:

We thank the reviewer again for their careful evaluation of the manuscript and positive opinion about its impact on the field of topological materials.

1) We agree that the presence of compressive strain is a crucial element for looking for Weyl physics in HgTe samples. The ability to introduce the compressive strain and change crystalline structure of HgTe has been proven previously in the 2019 Mahler et al. paper from our group (Ref. [16] in the article, 10.1103/PhysRevX.9.031034). In there, we verified by High resolution X-ray diffraction (HRXRD) analysis of 66 nm and 120 nm layers grown via the same MBE technology as the one used in this article. The paper reports, that the 66 nm layer is fully strained, with 0.3% compressive strain introduced by the lattice mismatch, while the 120 nm layer is partially relaxed, so that the strain is 25% less. The 72 nm layer studied in the present paper is close in thickness to the 66 nm layer from Ref. [16], i.e. is almost fully strained according to the lattice mismatch. HRXRD strain data are routinely obtained for every MBE-grown sample, including the one used in this study, so we exactly measured the amount of strain indicated in the manuscript (0.28%). We additionally reproduce the main results of HRXRD analysis from Supplementary Material of Ref. [16] below and additionally mention the strain measurement in the methods section.

[Figure Redacted]

2) Please find below the answers to individual remarks regarding the experimental evidence.

Reviewer's response:

The authors mention that their samples exhibit high mobility. Under such conditions, Shubnikov–de Haas (SdH) oscillations are typically readily observable in magnetoresistance measurements. The presence of SdH oscillations would constitute compelling evidence in identifying the Weyl nature of the material. Therefore, I strongly encourage the authors to include this crucial data in the supplementary materials to further strengthen the manuscript.

The authors' reply:

We thank the reviewer for inviting the discussion of quantum oscillations in Weyl semimetals. Indeed, the position of the intercept of SdH oscillations trend is sometimes interpreted as evidence for Weyl nature of the material, following, for example, the logic discussed in Ref. [28], which is used for high-density bulk materials.

However, this approach should be applied with extreme caution, as in many Weyl semimetals topological surface states emanate from the Weyl points and provide additional magnetoresistance oscillations and quantum Hall plateaus at high magnetic field. This is, for example, a well-established feature of (compressively strained) HgTe, Cd₂As₃ and Na₃Bi, which all belong to the same class of Dirac semimetals, in which the nodes do not appear at high symmetry points in the Brillouin zone. The Dirac nodes are split into Weyl nodes by BIA (bulk inversion asymmetry) and this splitting is obviously enhanced by the Zeeman effect when a magnetic field is applied.

Our previous study (Ref. [16]) focuses exactly on the topic of magnetoresistance and its relation to the surface states. For our low carrier density devices, we not only observe the SdH oscillations, but also the quantum Hall effect (QH) in the transverse resistance. We analyse the transverse conductivity, which is a combination of longitudinal (R_{xx}) and transverse (R_{xy}) resistances:

$$\sigma_{xy} = \frac{-R_{xy}}{R_{xy}^2 + (WR_{xx}/L)^2},$$

where W is the width and L is the length of the Hal bar. σ_{xy} provides means for even better understanding of the bandstructure compared to just SdH oscillations:

[Figure Redacted]

This 2D plot from Ref. [16] shows the derivative of σ_{xy} with gate voltage, yielding peaks at transitions between the Landau levels. The unusual plateau sequence is a mixture of contributions from the electronic topological surface states (black) and hole-like massive surface states (magenta). The observed quantum oscillations can therefore not be used to extract Berry curvature or confirm the Weyl band structure of the material, as they originate from the surface states.

In this plot, $U_{\text{gate}} = 4 \text{ V}$ corresponds to $n=13 \times 10^{11} \text{ cm}^{-2}$ while $U_{\text{gate}} = -4 \text{ V}$ corresponds to $n=-10 \times 10^{11} \text{ cm}^{-2}$ of hole conductance. In our new paper we operate in a smaller range around Weyl points, so that the carrier density varies between approximately $n=3.5 \times 10^{11} \text{ cm}^{-2}$ and $n=-2.2 \times 10^{11} \text{ cm}^{-2}$. The difference in applied gate voltages comes from the use of a different insulator (material and thickness) than in Ref. [16].

We added the corresponding references and discussion to the Supplementary information section 1, where the magnetoresistance data for perpendicular magnetic field is discussed.

Reviewer's response:

The authors repeatedly emphasize the high mobility of their samples and the proximity of the Fermi level to the Weyl point, yet no quantitative mobility data have been provided. In fact, such high mobility can readily give rise to the controversial current-jetting effect, which remains a significant concern in the field.

The authors' reply:

We thank the reviewer for pointing at the mobility and Fermi level position information that was missing in the manuscript. We have included the corresponding information into the manuscript (Methods section) and Supplementary Information. In this device we can control the total carrier density in the structure via the electrostatic gate between approximately $3.5 \times 10^{11} \text{ cm}^{-2}$ and $-2.2 \times 10^{11} \text{ cm}^{-2}$ of surface density (As already mentioned above, close to the Weyl points the transport is dominated by the surface states). According to our estimates via comparison with k dot p calculation in Fig.1b, these carrier densities correspond to Fermi level being moved in a ± 2 meV range around the Weyl nodes.

The mobility varies with the position of the Fermi level and is different for electronic and Hall states. We extract it from an analysis of the zero-field resistivity value. For a density $3.7 \times 10^{11} \text{ cm}^{-2}$ it reaches approx. $77 \times 10^3 \text{ cm}^2/(\text{Vs})$, while for the valence band at $V_g = -0.5\text{V}$ it is approx. $16 \times 10^3 \text{ cm}^2/(\text{Vs})$. Directly at the Weyl point the carrier density cannot be reliably extracted from the Hall slope, not allowing to obtain a precise mobility value.

We agree that current jetting can be a significant concern for bulk materials, particularly when soldering the metal leads directly on the bulk crystals. In our case, this phenomenon does not occur, as we can utilize the semiconductor technology developed over decades for our MBE-grown epilayers. The contacts are fabricated using standard lithography techniques in a Hall bar-like geometry, forming low-resistance (few tens of Ohms for the device studied in this paper, much smaller than the resistance of the studied structures, hundreds of Ohms, see Supplementary Fig. 3) bond pads and avoiding any possible nonuniformity. As a result, we observe the same negative magnetoresistance effect in both small structures, like the island studied in our paper and in large, macroscopic Hall bars studies in Ref. [16], which are hundreds of micrometres in size.

Reviewer's response:

In fact, the occurrence of weak antilocalization (WAL) and weak localization (WL) is not at all surprising in HgTe systems with strong spin-orbit coupling. Therefore, we remain unconvinced that the negative magnetoresistance observed below 2 T arises from the chiral anomaly, unless the authors can provide direct analytical evidence from SdH oscillations.

The authors' reply:

Indeed, these phenomena are routinely observed and well-studied for these materials. We would like to point the reviewers to the characteristic magnitude of WL and WAL, which stems from their physical origin. Both WL and WAL represent small corrections to the diffusive transport regime from quantum interference, that results in effective conductivity change on the order of one conductance quantum e^2/h . We observe a much stronger effect of the magnetic field, where the conductance in the in-plane field increases by approx. 30%. The presence of such a strong localization effect would imply strong disorder in the system, so that we are no longer operating in the diffusive transport regime. This cannot be the case because of the mobilities of our structures, so WL and WAL can be reliably excluded as a possible explanation. Moreover, at the

mobilities we have in our epilayers, any WL and WAL features are restricted to few tens mT width, as high mobility leads to narrow WAL signal, see also Ref. [A] below.

Reviewer's response:

In fact, as clearly shown in Supplementary Fig. 1 ($\Theta = 90^\circ$), the observed positive magnetoresistance is most likely attributable to weak antilocalization (WAL) arising from strong spin-orbit coupling, rather than to the positive magnetoresistance expected when the Fermi level is at the charge neutrality point. If the latter were the case, one would expect to see a pronounced parabolic increase in magnetoresistance at low fields for $\Theta=90^\circ$.

The authors' reply:

We strongly disagree with the interpretation of the data in Supplementary Fig. 1 ($\Theta = 90^\circ$) as WAL. If we look at the magnitude of the observed resistance increase with magnetic field, we see that it increases from 2 k Ω to 7 k Ω at 4 T. Weak antilocalization effects deliver small corrections to conductance of the order of e^2/h (typically even less than that, see for example Ref. [A]), and their suppression cannot result in such large increase of resistance with magnetic field, even in material with strong spin-orbit coupling.

The observed large magnetoresistance rather exactly matches expectations for a low-carrier-density system close to the Weyl point. As we discussed above, HgTe and other Weyl semimetals with similar crystalline structure possess surface states that reside close to the Weyl point, and whose contribution is observed in quantum hall and magnetoresistance close to the Weyl point. In our case, at -0.29V the carrier density is so low that

The reviewer probably expects even larger growth, as observed in some Weyl candidates, such as [21]. We note that this behaviour is only characteristic for 3D carriers and thus is not applicable to Weyl semimetal layers that also have surface states.

Reviewer's response:

I also recommend that the authors include relevant references when discussing these phenomena, in order to support their arguments appropriately.

The authors' reply:

We include the relevant references that were used in the main text in the discussion of thermal transport phenomena:

1. Ultraclean Weyl semimetals can be analysed based on a study by Andreev and Spivak [8]
2. The hydrodynamic transport regime is theoretically investigated by Lucas et al. [6] and by Andreev and Spivak [8]
3. The effects of electron-phonon coupling are discussed briefly in a review by Pekola and Karimi [39]
4. The elastic scattering regime is investigated by Andreev and Spivak [8].

Reviewer's response:

Finally, we regret to conclude that the manuscript still falls short of the standards required for publication in a high-profile journal such as Nature Communications. In the supplementary materials, we observed that for $\theta = 90^\circ$, oscillations appear in the magnetoresistance curve at fields above 2 T. Could the authors clarify the origin of these oscillations? Are they attributable to Shubnikov–de Haas (SdH) oscillations? If so, we strongly encourage the authors to extract and present key parameters such as the effective mass of the charge carriers, the precise location of the Fermi level, and the Berry curvature. The inclusion of such quantitative analyses would significantly enhance the credibility and impact of the manuscript.

The authors' reply:

We respectfully disagree with the evaluation that our manuscript falls short of the standards of this high-profile journal. We recognize that much of the criticism appears to stem from the absence of certain elements—Shubnikov–de Haas oscillations in perpendicular magnetic field and the extraction of Berry curvature from their phase—that are often expected in works on Weyl semimetals. These signatures are indeed frequently interpreted as hallmarks of Weyl physics, typically presented alongside negative magnetoresistance in in-plane field.

We note that this analysis is only applicable to bulk crystals with high carrier density and does not apply to layers of low-density material, as the one we investigate. In our case, no well-developed SdH oscillations can be observed when the Fermi level is located at the Weyl point due to the exceptionally low carrier density. This precisely matches our expectations for a Weyl semimetal, while the analysis of quantum oscillations at larger densities reliably assigns them to the surface states emanating from Weyl nodes, all studied in detail by Mahler et al. [16]. In low-density thin layer samples, oscillations with unusual phases rather emerge from a complex interplay of Landau levels, than reflecting intrinsic Berry curvature and bulk Weyl physics. Therefore, an attempt to interpret such oscillations in the conventional way would be fundamentally misleading. A copy of the corresponding paper has been included with our resubmission.

On the other hand, our devices do provide direct and reliable evidence that they are in the regime where Weyl physics is expected. By controlling the carrier density via the gate voltage, we can adjust the Fermi level position in the band structure with high precision. The entire range of our study spans only a few meV around the theoretically predicted Weyl point, as confirmed by our ability to tune continuously from electron-like to hole-like conduction. We identify the Weyl point at the gate voltage corresponding to the maximum negative magnetoresistance, which coincides precisely with the lowest carrier density. The existence of Weyl cones at this point is further confirmed by our $k \cdot p$ bandstructure calculations, which directly reflect the underlying crystalline symmetry.

We thank the reviewer for drawing attention to the oscillations observed in the island magnetoresistance for out-of-plane field orientation. However, these features do not appear to be related to Shubnikov–de Haas oscillations. They are only present in the small island, whereas in the larger channel, measured at the same Fermi level, we observe only a single broad oscillation associated with a Landau-level transition of the surface state—consistent with the behaviour shown in Ref. [16]. This indicates that the island oscillations most likely arise from magnetic field focusing effects on ballistic carrier dynamics, rather than reflecting the underlying band structure. This interpretation is fully consistent with our expectations: at such a low carrier density, well-developed SdH oscillations cannot occur at the accessible magnetic fields.

In summary, all our experimental observations align with the expected behaviour of a Weyl semimetal with low carrier density and significant surface-state contributions. Most importantly, the pronounced negative magnetoresistance observed only when the Fermi level is tuned near the Weyl point serves as the key transport signature of Weyl physics in our system. Building on this regime, our study of thermal conductance achieves its central goal—probing potential signatures of the gravitational anomaly in Weyl semimetals at low temperatures, where electron–phonon relaxation can be reliably excluded.

Reviewer #2 (Remarks to the Author):

All my concerns have been addressed. I recommend its publications.

The authors' reply:

We thank the reviewer for their evaluation of the manuscript.

A) Mühlbauer, M., et al. "One-dimensional weak antilocalization due to the Berry phase in HgTe wires." *Phys. Rev. Lett.* 112, 146803 – 2014.

RESPONSE TO THE REFEREE COMMENTS

Reviewer's response:

As for Current-jetting effect, please refers to Nature Reviews Physics 3, 394 (2021)

The authors' reply:

We thank the referee for drawing our attention to the review discussing current-jetting effects in high-mobility structures. In this context, we would like to highlight *Phys. Rev. X* **8**, 031002 (2018), which is cited in the review mentioned by the referee and identifies certain high-mobility Weyl semimetal candidates as systems where negative magnetoresistance can originate from current-jetting effects. In that work, a straightforward strategy to avoid current-jetting effects was proposed: fabricating thin, gate-tuneable films of the studied material to enable control over the Fermi-level position. This is precisely the approach implemented in our devices.

Importantly, current jetting occurs in structures where small, point-like contacts, directly glued onto a crystal, are used to inject current, leading to inhomogeneous current flow that can be strongly influenced by magnetic field [see Fig. 1c in *Nature Reviews Physics* **3**, 394 (2021)]. In contrast, our devices employ wide ohmic contacts fabricated using the standard lithography techniques of semiconductor technology, which are directly attached to the HgTe mesa (see Fig.2a of the manuscript). These diffusive contacts ensure a uniform current distribution within the structure, effectively eliminating magnetic-field-induced inhomogeneities that could give rise to current jetting.

Finally, our devices employ a thin 72 nm HgTe epilayer, for which the cyclotron radius at the magnetic fields used in our experiment is much larger than the layer thickness. Under these conditions, classical ballistic effects such as current jetting are strongly suppressed.

We added a reference to *Phys. Rev. X* **8**, 031002 (2018) to the Methods section in a sentence, discussing how our experimental approach allows to avoid current jetting effects.

Reviewer's response:

The authors state that the 30% increase in conductance under in-plane magnetic field is far too large to originate from weak localization (WL) or weak antilocalization (WAL), since these effects typically yield corrections on the order of e^2/h . However, this assertion is purely qualitative. Could the authors provide quantitative analysis or experimental evidence that definitively excludes WL/WAL as possible contributions? In other words, how do the authors demonstrate that a 30% change cannot arise—at least partially—from enhanced quantum-interference effects or multichannel localization in their system? Supporting data such as temperature dependence, angular dependence, or field-width analysis would be necessary to substantiate this claim.

The authors' reply:

Below we summarize the experimental evidence that clearly rules out quantum-interference mechanisms as the origin of the observed negative magnetoresistance and instead points to its band-structure origin:

1. Device dimensions

We investigated devices with vastly different geometric sizes (island of $6 \times 4.5 \mu\text{m}$ and a Hall bar of $1200 \times 200 \mu\text{m}$) and found that the relative magnitude of the negative magnetoresistance remains essentially unchanged. Moreover, the resistance does not show an exponential increase with length. These observations reliably exclude multimode localization as a possible explanation.

2. Temperature dependence

As discussed after the first review round, our devices exhibit only a weak temperature dependence, while the relative drop in resistance with magnetic field remains nearly constant up to 23 K (see Fig. 7.8 in the thesis of David Mahler, <https://nbn-resolving.org/urn:nbn:de:bvb:20-opus-253982>). This behaviour is inconsistent with both weak localization and multimode localization, which are expected to show a much stronger temperature dependence.

3. Angular dependence

The angular dependence of the magnetoresistance effect has been thoroughly characterized in *Phys. Rev. X* **9**, 031034 (2019) (Fig. 5) and is confirmed in our measurements (see Supplementary Section 1). The negative magnetoresistance appears only when the magnetic field is oriented in-plane and parallel to the current direction. This is opposite to the behavior expected for weak localization or multimode localization, where an out-of-plane field produces the strongest effect and an in-plane field has the weakest influence due to the minimal magnetic flux through electron trajectories.

4. Magnetic field width

Experimentally, we find that the negative magnetoresistance persists up to about 5 T, beyond which a positive magnetoresistance dominates. If this width were attributed to the suppression of quantum interference, it would correspond to a phase-coherence breakdown field $B_\phi \approx 5 \text{ T}$. Using $B_\phi = \hbar / (4el_\phi^2)$, this yields a phase-coherence length $l_\phi < 5 \text{ nm}$ — an unrealistically small value incompatible with the high mobility of our structures.

Reviewer's response:

Overall, the authors' explanations remain rather unconvincing. The goal of their work is clear: by applying external strain, they break the inversion symmetry of initially centrosymmetric HgTe, thereby transforming it into a Weyl semimetal. From this, they draw the following conclusion: "This finding indicates that, despite the unique electronic spectrum of Weyl semimetals, the mechanism governing heat transport in this system is the same as that for electrical transport, with no additional violations of conservation laws."

This is, without question, an important and potentially far-reaching conclusion. However, before reaching such a statement, the authors must first provide solid experimental demonstrations that (i) prior to strain the system was indeed a Dirac semimetal, supported by key evidence, and (ii) after strain, the transition to a Weyl semimetal has been directly confirmed by experiment. Only then can the final conclusion be considered physically convincing.

The authors' reply:

The ability to systematically control strain in HgTe layers and thereby tune their electronic band structure has been extensively demonstrated in previous studies over the past 30 years. Due to the inherent technological constraints of molecular beam epitaxy, truly unstrained HgTe films are rarely realized; most experimentally accessible samples are subject to either tensile or compressive strain.

Tensile-strained HgTe layers are a well-established realization of a three-dimensional topological insulator, in which the strain opens a bulk band gap and profoundly alters the transport properties. Representative studies include *Phys. Rev. Lett.* **106**, 126803 (2011); *Phys. Rev. X* **4**, 041045 (2014); *Nano Lett.* **21**, 9869–9874 (2021); and *Nano Lett.* **21**, 5195–5200 (2021).

Compressively strained layers, as investigated in *Phys. Rev. X* **9**, 031034 (2019) and in the present work, exhibit substantially different transport behavior. In contrast to the tensile case, these layers show no indication of a bulk gap, and when tuned close to charge neutrality, they display a negative magnetoresistance in in-plane magnetic fields. The classification of such structures as Weyl semimetals is ultimately supported by these distinct transport signatures, together with detailed band-structure calculations incorporating the experimentally determined strain values from the grown layers.

Moreover, the authors' replies often avoid the essential questions. For example, they argue that surface-state SdH oscillations interfere with bulk SdH signals, and therefore the usual SdH analysis cannot be applied. This argument is problematic. As they themselves emphasize, the strained samples are Weyl semimetals with extraordinarily high mobility ($77 \times 10^3 \text{cm}^2/\text{Vs}$) and with the Fermi energy very close to the Weyl point. Under such ideal conditions, the complete absence of SdH oscillations in magnetoresistance is highly counterintuitive and contradicts basic physical expectations, as well as previous studies (*Science* **350**, 413 (2015); *Phys. Rev. X* **5**, 031023 (2015); *Nat. Phys.* **14**, 1125 (2018)). Of course, there are known examples, such as Co_2MnGa , which ARPES has confirmed to be a topological semimetal yet still exhibits no SdH oscillations. However, in that material the absence of oscillations is attributed to an extremely high carrier density that produces large Landau-level spacing, making oscillatory magnetoresistance difficult to observe when the Fermi surface crosses Landau levels. In contrast, the experimental conditions reported for strained HgTe are optimal for observing topological features—low carrier density, high mobility, and a Fermi level precisely tuned to the Weyl point—yet no direct evidence of topological changes is observed. This is difficult to accept. Although the authors provide many explanations and checks, these do not directly relate to or convincingly demonstrate the topological transition itself.

The authors' reply:

We would like to clarify that SdH oscillations in perpendicular magnetic field are indeed observed in our devices when tuned slightly away from the Weyl point, as demonstrated in *Phys. Rev. X* **9**, 031034 (2019) and in Supplementary Fig. 7. Our point, however, is that these oscillations arise predominantly from the topological surface states rather than the bulk electrons, making it impossible to use them to extract the Berry phase of the bulk. Accessing SdH oscillations of the bulk would require suppressing the mobility of the surface states. Thus, there is no contradiction with basic physical principles or previous work; rather, we exercise caution in interpreting SdH oscillations as a definitive signature of bulk topology in this material. Directly at the Weyl point, the carrier density is simply too low for a distinct SdH oscillation pattern to be observable. Again, the band structure of HgTe-based structures has been explored extensively over the past 50 years, and the referee suggestions are at odds with a very well established chapter of semiconductor physics.

That said, I find the experimental method used to break inversion symmetry by strain application both technically precise and conceptually appealing. The quality of this experimental approach is impressive, and the study could stimulate productive debate among experts on this highly relevant problem. After all, research on transport signatures of the chiral anomaly in topological semimetals is still ongoing, and a clear consensus has not yet been reached. For this reason, while I remain skeptical about the claimed evidence of a Dirac-to-Weyl transition, I cautiously support publication of this work. Its controversial aspects are, in my view, part of what makes it interesting and may encourage further investigation and discussion in the field.

The authors' reply:

We thank the referee for their thoughtful assessment of our work and for recognizing the relevance of both the topic and our experimental approach. We fully agree that the research field of Weyl semimetals remains far from settled. One major challenge is that many of the proposed electronic-transport signatures used to identify Weyl physics are not uniquely attributable to the presence of Weyl nodes. The unusual phase offsets observed in SdH oscillations are often attributed to a nontrivial Berry phase, yet in practice they may arise from the coexistence of multiple conduction channels—such as topological surface states and bulk carriers—with distinct effective masses. Likewise, the large positive magnetoresistance observed in perpendicular magnetic fields is a well-known characteristic of nearly compensated semimetals and does not, by itself, provide conclusive evidence for Weyl physics.

This leaves the observation of a negative magnetoresistance, interpreted as a manifestation of the chiral anomaly, as one of the few remaining transport-based indicators. However, even this signature is not unique: in several high-mobility systems, similar behavior has been shown to result from classical current-jetting effects. Moreover, theoretical works have suggested that negative

magnetoresistance may appear in materials with any band crossings, not necessarily those hosting the Weyl nodes.

Thus, our present study is aimed not only at identifying the thermal transport properties of compressively strained HgTe but also exploring their potential as a probe of Weyl physics. This is particularly appealing, as the observation of quantum anomalies in a semiconductor platform is one of the major points that initially sparked interest in Weyl semimetals. We selected a material system, where the ability to adjust the band structure by strain engineering has already been demonstrated, while conventional fabrication methods are easily available to create complex structures with precise control of the Fermi level position. Unfortunately, our findings indicate that thermal transport does not exhibit distinct anomalies that could serve as a signature of Weyl points, thereby challenging current theories, that suggest quantum anomalies expected in particle physics to occur in the thermal transport in Weyl semimetals. The only anomaly our devices do demonstrate is the chiral anomaly in the electrical conductance, and they do so very convincingly.

RESPONSE TO REVIEWERS' COMMENTS

Reviewer #1 (Remarks to the Author):

Dear Editor,

After carefully reviewing this manuscript, I am impressed by the innovative and rigorous measurement techniques employed. The experimental data sufficiently support the core conclusion that "the low-temperature thermal transport of HgTe under stress still conforms to the Wiedemann-Franz law." However, the authors further assert that the compressively strained HgTe transforms into a Weyl fermion system, and that thermal transport does not violate the Wiedemann-Franz law. This claim lacks sufficient justification, and there are several specific issues that need clarification from the authors:

The authors' reply:

We thank the reviewer for carefully evaluating the manuscript and appreciate the constructive criticism. Please find our detailed response below.

1. Mechanism of Negative Magnetoresistance

As stated by the authors, negative magnetoresistance may arise from multiple mechanisms, and the positive magnetoresistance observed when the magnetic field exceeds 2T does conform to weak localization characteristics. However, the key question is: how do the authors rule out that the negative magnetoresistance in the magnetic field region below 2T is not caused by weak anti-localization effects? This requires more substantial experimental evidence for clarification.

The authors' reply:

Compressively strained HgTe exhibits a pronounced negative magnetoresistance (MR) when the magnetic and electric fields are aligned (i.e., $B \parallel E$). This negative magnetoresistance, however, is accompanied by positive MR features at both low and high magnetic field ranges. The positive magnetoresistance at low fields ($<0.5T$) is indeed attributed to the weak anti-localization (WAL) effect, a quantum interference phenomenon. The reviewer is probably referencing the weak localization (WL), another quantum interference phenomenon, that leads to resistance decrease with magnetic field and is commonly observed in combination with WAL.

Several arguments allow us to clearly distinguish the observed resistance decrease at intermediate fields (from 0.5T up to 5T) from WL:

First, due to the high mobility in our sample, the observed negative MR cannot be attributed to WL. Instead, at the expected very low magnetic fields, HgTe exhibits WAL (Ref. A), arising from strong spin-orbit coupling, on top of the negative MR background.

Reviewer's response:

The authors mention that their samples exhibit high mobility. Under such conditions, Shubnikov–de Haas (SdH) oscillations are typically readily observable in magnetoresistance measurements. The presence of SdH oscillations would constitute compelling evidence in identifying the Weyl nature of the material. Therefore, I strongly encourage the authors to include this crucial data in the supplementary materials to further strengthen the manuscript.

Second, WL is expected to be active when electrons can pick phase from magnetic fields. This is typically stronger for the field orientations, perpendicular to the current flow, due to the larger flux through electron trajectories. In contrast, in our system negative MR is only observed for in-plane magnetic field, parallel to the current flow, $B \parallel I$. For all other field directions, we observe positive MR (see the answer to the next question and newly added Supplementary Fig. 1 in the Supplementary Information). Specifically, for in-plane perpendicular we observe very weak positive MR, which is unlikely for WL.

Reviewer's response:

The authors repeatedly emphasize the high mobility of their samples and the proximity of the Fermi level to the Weyl point, yet no quantitative mobility data have been provided. In fact, such high mobility can readily give rise to the controversial current-jetting effect, which remains a significant concern in the field.

Third, the observed negative MR in compressed HgTe survives at high temperatures, while WL is suppressed quickly as the quantum coherence is destroyed. Below we present the data adopted from the thesis of David Mahler (<https://nbn-resolving.org/urn:nbn:de:bvb:20-opus-253982>, Fig.7.8), where the temperature dependence of magnetoresistance in compressively strained HgTe was measured under $B \parallel E$. The measurements were performed in our group on a different device with the layer stack and material properties closely matching those of the device discussed in this work. On top of the original figure panel b we added a mark pointing to the region of WAL.

[Figure redacted]

From Fig.7.8 (b), we observe that the sharp WAL feature at $B=0T$ quickly disappears with increasing temperature, confirming its origin from interference effects. In contrast, the pronounced negative MR at intermediate magnetic fields remains largely unchanged. This effectively rules out quantum interference mechanisms such as WL as the underlying cause. A noticeable change in the negative magnetoresistance is only observed at 23 K, likely due to enhanced phonon scattering at elevated temperatures.

Last, in our device we have a direct control over the position of the Fermi level in the band structure with an electrostatic gate. We observe prominent negative MR only when the Fermi level is tuned near the position of the Weyl points, which is verified via carrier density extracted from the out-of-plane magnetic field Hall measurement. This does not match the expectations for WL, where no direct relation to the band structure position of Fermi level is expected. At the same time, the band structure calculations based on $\mathbf{k}\cdot\mathbf{p}$ theory reveal the existence of linearly dispersing Weyl points in compressively strained HgTe, that should result in effect of chiral anomaly only for Fermi level close the Weyl points as detailed in Ref. [16].

These directional and energy-specific properties of our observations support the interpretation of the negative magnetoresistance as a signature of the chiral anomaly.

Reviewer's response:

In fact, the occurrence of weak antilocalization (WAL) and weak localization (WL) is not at all surprising in HgTe systems with strong spin-orbit coupling. Therefore, we remain unconvinced that the negative magnetoresistance observed below 2 T arises from the chiral anomaly, unless the authors can provide direct analytical evidence from SdH oscillations.

2. Completeness of Gate Voltage Regulation

The study achieves continuous control of the Fermi surface between the valence band and the conduction band through gate voltage. However, the data presented indicate that when $V_g = -0.29\text{V}$, the Fermi surface should be precisely located at the Weyl point, at which point a significant positive magnetoresistance effect should be observed under vertical magnetic field conditions. The absence of this critical experimental data weakens the conclusion regarding Weyl semimetals.

The authors' reply:

We have added measurements of the magnetoresistance of our device for out-of-plane magnetic field configuration to the Supplementary Materials. Indeed, for an out-of-plane field, when the Fermi level is tuned to the Weyl points, a pronounced positive magnetoresistance is observed. We also performed measurements with an in-plane magnetic field, perpendicular to the electric field, which results in very weak positive magnetoresistance. These observations indicate that the negative magnetoresistance is exclusive to the configuration where the magnetic field is aligned parallel to the electric field within the device, as expected for chiral anomaly. A comprehensive study on the angular dependence of magnetoresistance actually is available in our previous paper on compressively strained HgTe with a comparable layer stack and electronic properties (Ref. [16]), providing further insights into the role of chiral anomaly in the observed transport behavior.

Reviewer's response:

In fact, as clearly shown in Supplementary Fig. 1 ($\Theta = 90^\circ$), the observed positive magnetoresistance is most likely attributable to weak antilocalization (WAL) arising from strong spin-orbit coupling, rather than to the positive magnetoresistance expected when the Fermi level is at the charge neutrality point. If the latter were the case, one would expect to see a pronounced parabolic increase in magnetoresistance at low fields for $\Theta=90^\circ$.

3. Contradictory Phenomena in Magnetic Field Response

An analysis combining Figures 2b and 3d reveals that although a 2T magnetic field is a turning point for the dominance of chiral anomalies and weak localization, the Lorenz constant shows an unusually smooth variation within the range of 0-8T. Does this suggest that the thermal response across the entire magnetic field range (both $<2\text{T}$ and $>2\text{T}$) originates from trivial electronic states rather than Weyl fermions?

The authors' reply:

We thank the reviewer for their useful comment, that points directly to the core message of our study. The stability of the Lorenz ratio as a function of the magnetic fields demonstrates that the heat is transported via the same mechanism as the charge at all fields, and that it is transported via fermionic quasiparticles with charge $-e$. So, this does not mean that the thermal transport is

trivial, on the contrary, this indicates, that if the charge transport originates from the effect of the chiral anomaly, so does the heat transport.

However, theories of Weyl semimetals predict that their thermal conductance can be influenced by the so-called gravitational anomaly. Below we summarize how our observations relate to expectations across different transport regimes (we only consider those, where negative magnetoresistance from chiral anomaly should be present):

1. In an ultraclean Weyl semimetal, the chiral anomaly might affect the charge transport due to the anomalous charge pumping between the Weyl nodes, but less so the energy and heat transport. In that case one might expect $L < L_0$.
2. If inelastic e-e interactions are strong and dominate over all other elastic and inelastic scattering processes, a hydrodynamic transport regime might be realized, where electrons exhibit a form of collective motion, which should result in a substantial breakdown of the Wiedemann-Franz law, presumably with $L > L_0$. This is the regime where the gravitational anomaly effects might be observed in thermal transport.
3. Otherwise, if inelastic electron-phonon scattering dominates, which is expected for higher temperatures, electrons and phonons are strongly coupled and come to the local equilibrium. Thermal conductance then contains electron and phonon components, and the electronic part trivially follows $L=L_0$.
4. If elastic scattering dominates and the internode scattering rate is low, both charge and heat are transported via the same fermionic quasiparticles, and $L=L_0$ is expected, even in the presence of the anomaly.

Previous studies of thermal transport in Weyl systems were done in regime (3), due to the measurements being performed at high lattice temperatures, when electron-phonon scattering is strong. In contrast, we take special care to ensure that we are not operating in this regime by working at low lattice temperatures with fully electronic methods of heating and temperature detection, minimizing the effects of phonon drag and lattice heating. We also directly verify that the electron-phonon relaxation is small enough. However, we find, that under these conditions $L=L_0$, indicating that regime (2) is not realized and rather pointing to the regime (4), which is by no means trivial, but does not allow to experimentally detect the effect of gravitational anomaly.

Reviewer's response:

I also recommend that the authors include relevant references when discussing these phenomena, in order to support their arguments appropriately.

Considering the innovative thermal measurement techniques in this work and the cutting-edge nature of the research topic, I recommend that it is suitable for publication in Nature Communications after the above key evidence is supplemented, particularly providing conclusive experimental evidence that the material system possesses Weyl semimetal characteristics.

Reviewer's response:

Finally, we regret to conclude that the manuscript still falls short of the standards required for publication in a high-profile journal such as Nature Communications. In the supplementary materials, we observed that for $\theta = 90^\circ$, oscillations appear in the magnetoresistance curve at fields above 2 T. Could the authors clarify the origin of these oscillations? Are they attributable to Shubnikov–de Haas (SdH) oscillations? If so, we strongly encourage the authors to extract and present key parameters such as the effective mass of the charge carriers, the precise location of the Fermi level, and the Berry curvature. The inclusion of such quantitative analyses would significantly enhance the credibility and impact of the manuscript.

Reviewer #2 (Remarks to the Author):

This paper by A. Aravindnath, et al investigates the Wiedemann-Franz behavior in a Weyl semimetal based on compressively strained HgTe. By performing thermometry measurements at low temperatures using fully electronic methods, the research clearly identifies the Weyl regime and extracts the thermal conductance. The observed increase in thermal conductance with the applied magnetic field matches the increase in electrical conductance, adhering to the Wiedemann-Franz law. This indicates that, despite the unique electronic structure of Weyl semimetals, the thermal transport mechanism remains consistent with electrical transport, without violating any conservation laws. Overall, this work is interesting and novel, and I would be willing to recommend it for publication in Nature Communications after the authors address the following concerns:

As we know, in thermal conductance measurements, the heat flow or heat power Q flowing through the sample is a crucial and intuitive quantity, especially when studying scaling laws such as the Wiedemann-Franz law. However, the analysis and calculation of the heat power Q in the manuscript appear overly simplistic and cursory. I suggest that the authors provide a clearer and more detailed explanation of this heating power.

The authors' reply:

We thank the reviewer for their useful comments and appreciate their evaluation of our work. In the updated version of the manuscript, we reworked and expanded the discussion of the extraction of the heat flux, flowing through the sample and added extra details to support our analysis. We added extra subsection to the "Methods" that explains our approach to finding the heat flux.

Reviewer #3 (Remarks to the Author):

This paper mainly studies the thermal transport properties of compressively strained HgTe layer. The Fermi level can be tuned by changing the gate voltage. The authors found that even though Weyl fermions are located at the Fermi level and dominate transport properties, the Wiedemann-Franz law still holds, indicating that Weyl fermions do not cause distinct physics in thermal transport. In my opinion, this paper does not meet the high standards of Nature Communications, and other journals such as Communications Physics may be more suitable.

(1) The innovation of the paper is not sufficient to be published in NC. I don't think the results of this article are very interesting.

The authors' reply:

We respectfully disagree with the reviewer on this point. In our view, the present work makes a meaningful contribution to the study of Weyl semimetals. The question of whether the Wiedemann–Franz law is violated—and whether hydrodynamic electron thermal conductance can be detected—is central to experimental efforts aimed at observing the solid-state analogue of the gravitational anomaly in these systems. In fact, the original proposal that Weyl semimetals could serve as a platform for such studies has been a key motivation for much of the thermal transport research in this field.

However, a reliable experimental confirmation of nontrivial heat transport mechanisms in Weyl semimetals remains elusive. This is due in part to the challenge of reaching a regime where electron–phonon scattering is weak, and in part to the intrinsic difficulty of isolating the electronic contribution to thermal conductance. In this work, we present an experiment that explicitly addresses both issues with care—yet we still do not observe the expected thermal anomaly. We believe this result provides important insight into the nature of thermal transport in this class of materials.

(2) In the “results” section, the author spent a considerable amount of space describing how the experiment was conducted, which should be included in the “methods” section. The main text should mainly describe and discuss the experimental results.

The authors' reply:

We acknowledge that the details of Johnson-Nyquist thermometry might not be interesting for every reader of the manuscript. Yet in our case the choice of the method of thermal conductance measurement is strongly connected to the observed physics: we want to operate in a regime where the phonon contribution is small and electron-phonon coupling is weak enough to be able to resolve the specific mechanisms of electronic transport. In the updated version of the manuscript, we moved some less important description of the measurement into the methods section, while keeping the necessary details to support this message.

(3) The data is not abundant and the quality is not high.

The authors' reply:

We would like to draw the reviewer's attention to the details of our study, in particular, to the research method we used. High precision Johnson-Nyquist thermometry is a very time-consuming experimental technique, requiring prolonged measurements to achieve high precision due to the statistical nature of the signal. The value that one should look at to understand the final precision is the relative error $\Delta T/T$. In our case we achieved ΔT of approx. 5mK for $T \sim 1.3\text{K}$ and 95% confidence interval. By this parameter our measurement is on par with the best practices in the field.

Additionally, to improve this parameter by n times, the measurement time needs to be multiplied by n^2 . This can potentially result in enormous measurement times, that don't necessary improve the analytic potential of the data. A study that results in high precision thermal transport coefficients, however, still requires several months of continuous measurements.

The choice of the research method is not our whim - the fragility of electronic thermal transport with respect to electron-phonon interactions and the need to accurately control heat flows in the system simply does not allow the use of other approaches, which we discuss in detail in the text of the article.

Given all this, we believe that the quantity of data and their quality are sufficient to reliably support the main conclusions of our article and follow the best practices in the field.

- A) Mühlbauer, M., et al. "One-dimensional weak antilocalization due to the Berry phase in HgTe wires." *Phys. Rev. Lett.* 112, 146803 – 2014.

RESPONSE TO THE REFEREE COMMENTS

Reviewer #1 (Remarks to the Author):

This article elaborates on the research conducted to modulate the topology of HgTe by precisely applying external stress to alter its crystal structure. This work represents significant advancement and importance in the field of topological materials. However, the determination of whether the modulation is successful must be approached rigorously, primarily based on the following two criteria: 1) Whether the crystal structure has indeed changed. This serves as the foundation for verifying the modulation effect. 2) Whether the key experimental evidence supporting the two types of topological properties is sufficient. This is crucial for ensuring the reliability of the research conclusions.

As these critical conditions have not yet been fully verified, it is currently not possible to scientifically conclude whether this research has achieved its intended goals.

The authors' reply:

We thank the reviewer again for their careful evaluation of the manuscript and positive opinion about its impact on the field of topological materials.

1) We agree that the presence of compressive strain is a crucial element for looking for Weyl physics in HgTe samples. The ability to introduce the compressive strain and change crystalline structure of HgTe has been proven previously in the 2019 Mahler et al. paper from our group (Ref. [16] in the article, 10.1103/PhysRevX.9.031034). In there, we verified by High resolution X-ray diffraction (HRXRD) analysis of 66 nm and 120 nm layers grown via the same MBE technology as the one used in this article. The paper reports, that the 66 nm layer is fully strained, with 0.3% compressive strain introduced by the lattice mismatch, while the 120 nm layer is partially relaxed, so that the strain is 25% less. The 72 nm layer studied in the present paper is close in thickness to the 66 nm layer from Ref. [16], i.e. is almost fully strained according to the lattice mismatch. HRXRD strain data are routinely obtained for every MBE-grown sample, including the one used in this study, so we exactly measured the amount of strain indicated in the manuscript (0.28%). We additionally reproduce the main results of HRXRD analysis from Supplementary Material of Ref. [16] below and additionally mention the strain measurement in the methods section.

[Figure redacted]

2) Please find below the answers to individual remarks regarding the experimental evidence.

Reviewer's response:

I accept this interpretation.

Reviewer's response:

The authors mention that their samples exhibit high mobility. Under such conditions, Shubnikov–de Haas (SdH) oscillations are typically readily observable in magnetoresistance measurements. The presence of SdH oscillations would constitute compelling evidence in identifying the Weyl nature of the material. Therefore, I strongly encourage the authors to include this crucial data in the supplementary materials to further strengthen the manuscript.

The authors' reply:

We thank the reviewer for inviting the discussion of quantum oscillations in Weyl semimetals. Indeed, the position of the intercept of SdH oscillations trend is sometimes interpreted as evidence for Weyl nature of the material, following, for example, the logic discussed in Ref. [28], which is used for high-density bulk materials.

However, this approach should be applied with extreme caution, as in many Weyl semimetals topological surface states emanate from the Weyl points and provide additional magnetoresistance oscillations and quantum Hall plateaus at high magnetic field. This is, for example, a well-established feature of (compressively strained) HgTe, Cd₂As₃ and Na₃Bi, which all belong to the same class of Dirac semimetals, in which the nodes do not appear at high symmetry points in the Brillouin zone. The Dirac nodes are split into Weyl nodes by BIA (bulk inversion asymmetry) and this splitting is obviously enhanced by the Zeeman effect when a magnetic field is applied.

Our previous study (Ref. [16]) focuses exactly on the topic of magnetoresistance and its relation to the surface states. For our low carrier density devices, we not only observe the SdH oscillations, but also the quantum Hall effect (QH) in the transverse resistance. We analyse the transverse conductivity, which is a combination of longitudinal (R_{xx}) and transverse (R_{xy}) resistances:

$$\sigma_{xy} = \frac{-R_{xy}}{R_{xy}^2 + (WR_{xx}/L)^2},$$

where W is the width and L is the length of the Hal bar. σ_{xy} provides means for even better understanding of the bandstructure compared to just SdH oscillations:

[Figure redacted]

This 2D plot from Ref. [16] shows the derivative of σ_{xy} with gate voltage, yielding peaks at transitions between the Landau levels. The unusual plateau sequence is a mixture of contributions from the electronic topological surface states (black) and hole-like massive surface states (magenta). The observed quantum oscillations can therefore not be used to extract Berry curvature or confirm the Weyl band structure of the material, as they originate from the surface states.

In this plot, $U_{\text{gate}} = 4 \text{ V}$ corresponds to $n=13 \times 10^{11} \text{ cm}^{-2}$ while $U_{\text{gate}} = -4 \text{ V}$ corresponds to $n=-10 \times 10^{11} \text{ cm}^{-2}$ of hole conductance. In our new paper we operate in a smaller range around Weyl points, so that the carrier density varies between approximately $n=3.5 \times 10^{11} \text{ cm}^{-2}$ and $n=-2.2 \times 10^{11} \text{ cm}^{-2}$. The difference in applied gate voltages comes from the use of a different insulator (material and thickness) than in Ref. [16].

We added the corresponding references and discussion to the Supplementary information section 1, where the magnetoresistance data for perpendicular magnetic field is discussed.

Reviewer's response:

I accept this interpretation.

Reviewer's response:

The authors repeatedly emphasize the high mobility of their samples and the proximity of the Fermi level to the Weyl point, yet no quantitative mobility data have been provided. In fact, such high mobility can readily give rise to the controversial current-jetting effect, which remains a significant concern in the field.

The authors' reply:

We thank the reviewer for pointing at the mobility and Fermi level position information that was missing in the manuscript. We have included the corresponding information into the manuscript (Methods section) and Supplementary Information. In this device we can control the total carrier density in the structure via the electrostatic gate between approximately $3.5 \times 10^{11} \text{ cm}^{-2}$ and $-2.2 \times 10^{11} \text{ cm}^{-2}$ of surface density (As already mentioned above, close to the Weyl points the transport is dominated by the surface states). According to our estimates via comparison with $k \cdot p$ calculation in Fig.1b, these carrier densities correspond to Fermi level being moved in a ± 2 meV range around the Weyl nodes.

The mobility varies with the position of the Fermi level and is different for electronic and Hall states. We extract it from an analysis of the zero-field resistivity value. For a density $3.7 \times 10^{11} \text{ cm}^{-2}$ it reaches approx. $77 \times 10^3 \text{ cm}^2/(\text{Vs})$, while for the valence band at $V_g = -0.5\text{V}$ it is approx. $16 \times 10^3 \text{ cm}^2/(\text{Vs})$. Directly at the Weyl point the carrier density cannot be reliably extracted from the Hall slope, not allowing to obtain a precise mobility value.

We agree that current jetting can be a significant concern for bulk materials, particularly when soldering the metal leads directly on the bulk crystals. In our case, this phenomenon does not occur, as we can utilize the semiconductor technology developed over decades for our MBE-grown epilayers. The contacts are fabricated using standard lithography techniques in a Hall bar-like geometry, forming low-resistance (few tens of Ohms for the device studied in this paper, much smaller than the resistance of the studied structures, hundreds of Ohms, see Supplementary Fig. 3) bond pads and avoiding any possible nonuniformity. As a result, we observe the same negative magnetoresistance effect in both small structures, like the island studied in our paper and in large, macroscopic Hall bars studies in Ref. [16], which are hundreds of micrometres in size.

Reviewer's response:

As for Current-jetting effect, please refers to Nature Reviews Physics 3, 394 (2021)

Reviewer's response:

In fact, the occurrence of weak antilocalization (WAL) and weak localization (WL) is not at all surprising in HgTe systems with strong spin-orbit coupling. Therefore, we remain unconvinced that the negative magnetoresistance observed below 2 T arises from the chiral anomaly, unless the authors can provide direct analytical evidence from SdH oscillations.

The authors' reply:

Indeed, these phenomena are routinely observed and well-studied for these materials. We would like to point the reviewers to the characteristic magnitude of WL and WAL, which stems from their physical origin. Both WL and WAL represent small corrections to the diffusive transport regime from quantum interference, that results in effective conductivity change on the order of one conductance quantum e^2/h . We observe a much stronger effect of the magnetic field, where the conductance in the in-plane field increases by approx. 30%. The presence of such a strong localization effect would imply strong disorder in the system, so that we are no longer operating in the diffusive transport regime. This cannot be the case because of the mobilities of our structures, so WL and WAL can be reliably excluded as a possible explanation. Moreover, at the mobilities we have in our epilayers, any WL and WAL features are restricted to few tens mT width, as high mobility leads to narrow WAL signal, see also Ref. [A] below.

Reviewer's response:

The authors state that the 30% increase in conductance under in-plane magnetic field is far too large to originate from weak localization (WL) or weak antilocalization (WAL), since these effects typically yield corrections on the order of e^2/h . However, this assertion is purely qualitative. Could the authors provide quantitative analysis or experimental evidence that definitively excludes WL/WAL as possible contributions? In other words, how do the authors demonstrate that a 30% change cannot arise—at least partially—from enhanced quantum-interference effects or multichannel localization in their system? Supporting data such as temperature dependence, angular dependence, or field-width analysis would be necessary to substantiate this claim.

Reviewer's response:

In fact, as clearly shown in Supplementary Fig. 1 (Theta = 90°), the observed positive magnetoresistance is most likely attributable to weak antilocalization (WAL) arising from strong spin-orbit coupling, rather than to the positive magnetoresistance expected when the Fermi level is at the charge neutrality point. If the latter were the case, one would expect to see a pronounced parabolic increase in magnetoresistance at low fields for Theta=90°.

The authors' reply:

We strongly disagree with the interpretation of the data in Supplementary Fig. 1 (Theta = 90°) as WAL. If we look at the magnitude of the observed resistance increase with magnetic field, we see that it increases from 2 kΩ to 7 kΩ at 4 T. Weak antilocalization effects deliver small corrections to conductance of the order of e^2/h (typically even less than that, see for example Ref. [A]), and their suppression cannot result in such large increase of resistance with magnetic field, even in material with strong spin-orbit coupling.

The observed large magnetoresistance rather exactly matches expectations for a low-carrier-density system close to the Weyl point. As we discussed above, HgTe and other Weyl semimetals with similar crystalline structure possess surface states that reside close to the Weyl point, and whose contribution is observed in quantum hall and magnetoresistance close to the Weyl point. In our case, at -0.29V the carrier density is so low that

The reviewer probably expects even larger growth, as observed in some Weyl candidates, such as [21]. We note that this behaviour is only characteristic for 3D carriers and thus is not applicable to Weyl semimetal layers that also have surface states.

Reviewer's response:

I partially accept this interpretation.

Reviewer's response:

I also recommend that the authors include relevant references when discussing these phenomena, in order to support their arguments appropriately.

The authors' reply:

We include the relevant references that were used in the main text in the discussion of thermal transport phenomena:

1. Ultraclean Weyl semimetals can be analysed based on a study by Andreev and Spivak [8]
2. The hydrodynamic transport regime is theoretically investigated by Lucas et al. [6] and by Andreev and Spivak [8]
3. The effects of electron-phonon coupling are discussed briefly in a review by Pekola and Karimi [39]
4. The elastic scattering regime is investigated by Andreev and Spivak [8].

Reviewer's response:

I accept this interpretation.

Reviewer's response:

Finally, we regret to conclude that the manuscript still falls short of the standards required for publication in a high-profile journal such as Nature Communications. In the supplementary materials, we observed that for $\theta = 90^\circ$, oscillations appear in the magnetoresistance curve at fields above 2 T. Could the authors clarify the origin of these oscillations? Are they attributable to Shubnikov–de Haas (SdH) oscillations? If so, we strongly encourage the authors to extract and present key parameters such as the effective mass of the charge carriers, the precise location of the Fermi level, and the Berry curvature. The inclusion of such quantitative analyses would significantly enhance the credibility and impact of the manuscript.

The authors' reply:

We respectfully disagree with the evaluation that our manuscript falls short of the standards of this high-profile journal. We recognize that much of the criticism appears to stem from the absence of certain elements—Shubnikov–de Haas oscillations in perpendicular magnetic field and the extraction of Berry curvature from their phase—that are often expected in works on Weyl semimetals. These signatures are indeed frequently interpreted as hallmarks of Weyl physics, typically presented alongside negative magnetoresistance in in-plane field.

We note that this analysis is only applicable to bulk crystals with high carrier density and does not apply to layers of low-density material, as the one we investigate. In our case, no well-developed SdH oscillations can be observed when the Fermi level is located at the Weyl point due to the exceptionally low carrier density. This precisely matches our expectations for a Weyl semimetal, while the analysis of quantum oscillations at larger densities reliably assigns them to the surface states emanating from Weyl nodes, all studied in detail by Mahler et al. [16]. In low-density thin layer samples, oscillations with unusual phases rather emerge from a complex interplay of Landau levels, than reflecting intrinsic Berry curvature and bulk Weyl physics. Therefore, an attempt to interpret such oscillations in the conventional way would be fundamentally misleading. A copy of the corresponding paper has been included with our resubmission.

On the other hand, our devices do provide direct and reliable evidence that they are in the regime where Weyl physics is expected. By controlling the carrier density via the gate voltage, we can adjust the Fermi level position in the band structure with high precision. The entire range of our study spans only a few meV around the theoretically predicted Weyl point, as confirmed by our ability to tune continuously from electron-like to hole-like conduction. We identify the Weyl point at the gate voltage corresponding to the maximum negative magnetoresistance, which coincides precisely with the lowest carrier density. The existence of Weyl cones at this point is further confirmed by our $k \cdot p$ bandstructure calculations, which directly reflect the underlying crystalline symmetry.

We thank the reviewer for drawing attention to the oscillations observed in the island magnetoresistance for out-of-plane field orientation. However, these features do not appear to be related to Shubnikov–de Haas oscillations. They are only present in the small island, whereas in the larger channel, measured at the same Fermi level, we observe only a single broad oscillation associated with a Landau-level transition of the surface state—consistent with the behaviour shown in Ref. [16]. This indicates that the island oscillations most likely arise from magnetic field focusing effects on ballistic carrier dynamics, rather than reflecting the underlying band structure.

This interpretation is fully consistent with our expectations: at such a low carrier density, well-developed SdH oscillations cannot occur at the accessible magnetic fields.

In summary, all our experimental observations align with the expected behaviour of a Weyl semimetal with low carrier density and significant surface-state contributions. Most importantly, the pronounced negative magnetoresistance observed only when the Fermi level is tuned near the Weyl point serves as the key transport signature of Weyl physics in our system. Building on this regime, our study of thermal conductance achieves its central goal—probing potential signatures of the gravitational anomaly in Weyl semimetals at low temperatures, where electron–phonon relaxation can be reliably excluded.

Reviewer's response:

Overall, the authors' explanations remain rather unconvincing. The goal of their work is clear: by applying external strain, they break the inversion symmetry of initially centrosymmetric HgTe, thereby transforming it into a Weyl semimetal. From this, they draw the following conclusion: "This finding indicates that, despite the unique electronic spectrum of Weyl semimetals, the mechanism governing heat transport in this system is the same as that for electrical transport, with no additional violations of conservation laws."

This is, without question, an important and potentially far-reaching conclusion. However, before reaching such a statement, the authors must first provide solid experimental demonstrations that (i) prior to strain the system was indeed a Dirac semimetal, supported by key evidence, and (ii) after strain, the transition to a Weyl semimetal has been directly confirmed by experiment. Only then can the final conclusion be considered physically convincing.

Moreover, the authors' replies often avoid the essential questions. For example, they argue that surface-state SdH oscillations interfere with bulk SdH signals, and therefore the usual SdH analysis cannot be applied. This argument is problematic. As they themselves emphasize, the strained samples are Weyl semimetals with extraordinarily high mobility ($77 \times 10^3 \text{ cm}^2/\text{Vs}$) and with the Fermi energy very close to the Weyl point. Under such ideal conditions, the complete absence of SdH oscillations in magnetoresistance is highly counterintuitive and contradicts basic physical expectations, as well as previous studies (Science 350, 413 (2015); Phys. Rev. X 5, 031023 (2015); Nat. Phys. 14, 1125 (2018)). Of course, there are known examples, such as Co_2MnGa , which ARPES has confirmed to be a topological semimetal yet still exhibits no SdH oscillations. However, in that material the absence of oscillations is attributed to an extremely high carrier density that produces large Landau-level spacing, making oscillatory magnetoresistance difficult to observe when the Fermi surface crosses Landau levels. In contrast, the experimental conditions reported for strained HgTe are optimal for observing topological features—low carrier density, high mobility, and a Fermi level precisely tuned to the Weyl point—yet no direct evidence of topological changes is observed. This is difficult to accept. Although the authors provide many explanations and checks, these do not directly relate to or convincingly demonstrate the topological transition itself.

That said, I find the experimental method used to break inversion symmetry by strain application both technically precise and conceptually appealing. The quality of this experimental approach is impressive, and the study could stimulate productive debate among experts on this highly relevant problem. After all,

research on transport signatures of the chiral anomaly in topological semimetals is still ongoing, and a clear consensus has not yet been reached. For this reason, while I remain skeptical about the claimed evidence of a Dirac-to-Weyl transition, I cautiously support publication of this work. Its controversial aspects are, in my view, part of what makes it interesting and may encourage further investigation and discussion in the field.

Reviewer #2 (Remarks to the Author):

All my concerns have been addressed. I recommend its publications.

The authors' reply:

We thank the reviewer for their evaluation of the manuscript.

A) Mühlbauer, M., et al. "One-dimensional weak antilocalization due to the Berry phase in HgTe wires." *Phys. Rev. Lett.* 112, 146803 – 2014.